# Contextualized Messages Boost Graph Representations

**Brian Godwin Lim**                                         *lim.brian_godwin_sy.la6@is.naist.jp*
*Nara Institute of Science and Technology*

**Galvin Brice Lim**                                             *galvin_lim@dlsu.edu.ph*
*De La Salle University - Manila*

**Renzo Roel Tan**                                               *rr.tan@is.naist.jp*
*Nara Institute of Science and Technology & Ateneo de Manila University*

**Kazushi Ikeda**                                               *kazushi@is.naist.jp*
*Nara Institute of Science and Technology*

**Reviewed on OpenReview:** *https://openreview.net/forum?id=sXr1fRjs1N*

## Abstract

Graph neural networks (GNNs) have gained significant attention in recent years for their ability to process data that may be represented as graphs. This has prompted several studies to explore their representational capability based on the graph isomorphism task. Notably, these works inherently assume a countable node feature representation, potentially limiting their applicability. Interestingly, only a few study GNNs with uncountable node feature representation. In the paper, a new perspective on the representational capability of GNNs is investigated across all levels—node-level, neighborhood-level, and graph-level—when the space of node feature representation is uncountable. Specifically, the injective and metric requirements of previous works are *softly* relaxed by employing a *pseudometric* distance on the space of input to create a *soft-injective* function such that distinct inputs may produce *similar* outputs if and only if the *pseudometric* deems the inputs to be sufficiently *similar* on some representation. As a consequence, a simple and computationally efficient *soft-isomorphic* relational graph convolution network (SIR-GCN) that emphasizes the contextualized transformation of neighborhood feature representations via *anisotropic* and *dynamic* message functions is proposed. Furthermore, a mathematical discussion on the relationship between SIR-GCN and key GNNs in literature is laid out to put the contribution into context, establishing SIR-GCN as a generalization of classical GNN methodologies. To close, experiments on synthetic and benchmark datasets demonstrate the relative superiority of SIR-GCN, outperforming comparable models in node and graph property prediction tasks.

## 1 Introduction

Graph neural networks (GNNs) constitute a class of deep learning models designed to process data that may be represented as graphs. These models are well-suited for node, edge, and graph property prediction tasks across various domains, including social networks, molecular graphs, and biological networks, among others (Hu et al., 2020a; Dwivedi et al., 2023). In the literature, GNNs predominantly follow the message-passing scheme wherein each node aggregates the feature representation of its neighbors and combines them to create an updated node feature representation (Gilmer et al., 2017; Xu et al., 2018; 2019). This allows the model to encapsulate both the network structure and the broader node contexts. Moreover, a graph readout function is employed to pool the node feature representations of a graph and create an aggregated representation for the entire graph (Ying et al., 2018; Murphy et al., 2019; Xu et al., 2019).

Among the classical GNNs in literature include the graph convolution network (GCN) (Kipf & Welling, 2017), graph sample and aggregate (GraphSAGE) (Hamilton et al., 2017), graph attention network (GAT)

(Veličković et al., 2018; Brody et al., 2022), and graph isomorphism network (GIN) (Xu et al., 2019) which largely fall under the message-passing neural network (MPNN) (Gilmer et al., 2017) framework. These models have gained popularity due to their simplicity and remarkable performance across various applications (Hu et al., 2020a; Dwivedi et al., 2023). Improvements of these foundational models are also constantly proposed to achieve state-of-the-art performance (Wang et al., 2019b; Bodnar et al., 2021; Bouritsas et al., 2023).

Notably, advances in GNN have mainly been driven by heuristics and empirical results. Nonetheless, several studies have recently begun exploring the representational capability of GNNs (Garg et al., 2020; Sato et al., 2021; Azizian & Lelarge, 2021; Bodnar et al., 2021; Böker et al., 2023). Most of these works analyzed GNNs in relation to the graph isomorphism task. In particular, Xu et al. (2019) was among the first to lay the foundations for creating a maximally expressive GNN based on the Weisfeiler-Leman (WL) graph isomorphism test (Weisfeiler & Leman, 1968). Subsequent works build upon their results by considering extensions to the original 1-WL test. Crucially, the theoretical results of these works only hold with countable node feature representation which potentially limits their applicability. Meanwhile, Corso et al. (2020) proposed using multiple aggregators to create powerful GNNs when the space of node feature representation is uncountable. Interestingly, there has been no significant theoretical progress since this work.

This paper presents a new perspective on the representational capability of GNNs when the space of node feature representation is uncountable. Specifically, the key idea is to define a *pseudometric* distance on the space of input to create a *soft-injective* function such that distinct inputs may produce *similar* outputs if and only if the distance between the inputs is sufficiently small on some representation. This framework is comprehensively analyzed across all levels—node-level, neighborhood-level, and graph-level. From the theoretical results, a simple and computationally efficient *soft-isomorphic* relational graph convolution network (SIR-GCN) which emphasizes the contextualized transformation of neighborhood feature representations using *anisotropic* and *dynamic* message functions is proposed. This is further accompanied by a discussion on the mathematical relationship between SIR-GCN and key GNNs in literature to underscore the contribution and distinctive advantages of the proposed model. Finally, experiments on synthetic and benchmark datasets in node and graph property prediction tasks are performed to highlight the expressivity of SIR-GCN, positioning the model as a strong candidate for practical GNN applications.

## 2 Graph neural networks

Let $\mathcal{G} = (\mathcal{V}_{\mathcal{G}}, \mathcal{E}_{\mathcal{G}})$ be a graph and $\mathcal{N}_{\mathcal{G}}(u) \subseteq \mathcal{V}_{\mathcal{G}}$ be the set of nodes adjacent to node $u \in \mathcal{V}_{\mathcal{G}}$. The subscript $\mathcal{G}$ will be omitted whenever the context is clear. Suppose $\mathcal{H}$ is the space of node feature representation, henceforth feature, and $\boldsymbol{h_u} \in \mathcal{H}$ is the feature of node $u$. A GNN following the message-passing scheme can be expressed mathematically as

$$
\begin{aligned}
\boldsymbol{H_u} &:= \{\!\!\{\boldsymbol{h_v} : v \in \mathcal{N}_{\mathcal{G}}(u)\}\!\!\} \\
\boldsymbol{a_u} &:= \mathrm{AGG}\left(\boldsymbol{H_u}\right) \\
\boldsymbol{h_u^*} &:= \mathrm{COMB}\left(\boldsymbol{h_u}, \boldsymbol{a_u}\right),
\end{aligned}
\tag{1}
$$

where AGG and COMB are some aggregation and combination strategies, respectively, $\boldsymbol{H_u}$ is the *multiset* (Xu et al., 2019) of neighborhood features for node $u$, $\boldsymbol{a_u}$ is the aggregated neighborhood feature for node $u$, and $\boldsymbol{h_u^*}$ is the updated feature for node $u$. Since AGG takes arbitrary-sized *multisets* of neighborhood features as input and transforms them into a single feature, it may be considered a hash function. Hence, aggregation and hash functions shall be used interchangeably throughout the paper.

**Related works** When $\mathcal{H}$ is countable, Xu et al. (2019) showed that there exists a function $f : \mathcal{H} \to \mathcal{S}$ such that the aggregation or hash function

$$
F\left(\boldsymbol{H}\right) := \sum_{\boldsymbol{h} \in \boldsymbol{H}} f\left(\boldsymbol{h}\right)
\tag{2}
$$

is injective or unique in the embedded feature space $\mathcal{S}$ for each *multiset* of neighborhood features $\boldsymbol{H}$ of bounded size. This result forms the theoretical basis of GIN.

Meanwhile, the result above no longer holds when $\mathcal{H}$ is uncountable. In this setting, Corso et al. (2020) proved that if $\bigoplus$ comprises multiple aggregators (*e.g.*, mean, standard deviation, max, and min), the hash function

$$M\left(\boldsymbol{H}\right) \coloneqq \bigoplus_{\boldsymbol{h} \in \boldsymbol{H}} m\left(\boldsymbol{h}\right) \tag{3}$$

produces a unique output for every $\boldsymbol{H}$ of bounded size. This finding provides the foundation for the principal neighborhood aggregation (PNA) (Corso et al., 2020). Notably, for this result to hold theoretically, the number of aggregators in $\bigoplus$ must scale with the size of the *multiset* of neighborhood features $\boldsymbol{H}$ which may be impractical for large and dense graphs.

## 3 *Soft-injective* functions

While injective functions and metrics are necessary for tasks requiring graph isomorphism to ensure a unique mapping in the embedded feature space, many practical applications of GNN often do not require such strict constraints. For instance, in node classification tasks, GNNs must produce identical outputs for some distinct nodes. Thus, motivated by Xu et al. (2019) and Corso et al. (2020), this paper *softly* relaxes the injective and metric requirements within the MPNN framework by employing *pseudometrics* and *soft-injective* functions.

**Definition 1** (*Pseudometric*). *Let $\mathcal{H}$ be a non-empty set. A function $d : \mathcal{H} \times \mathcal{H} \to \mathbb{R}_{\geq 0}$ is a pseudometric on $\mathcal{H}$ if the following holds for all $\boldsymbol{h}^{(1)}, \boldsymbol{h}^{(2)}, \boldsymbol{h}^{(3)} \in \mathcal{H}$:*

*1. $d\left(\boldsymbol{h}^{(1)}, \boldsymbol{h}^{(1)}\right) = 0$;*

*2. $d\left(\boldsymbol{h}^{(1)}, \boldsymbol{h}^{(2)}\right) = d\left(\boldsymbol{h}^{(2)}, \boldsymbol{h}^{(1)}\right)$; and*

*3. $d\left(\boldsymbol{h}^{(1)}, \boldsymbol{h}^{(3)}\right) \leq d\left(\boldsymbol{h}^{(1)}, \boldsymbol{h}^{(2)}\right) + d\left(\boldsymbol{h}^{(2)}, \boldsymbol{h}^{(3)}\right)$.*

Note that for a metric $d$, the first condition is replaced with $d\left(\boldsymbol{h}^{(1)}, \boldsymbol{h}^{(2)}\right) = 0 \iff \boldsymbol{h}^{(1)} = \boldsymbol{h}^{(2)}$, ensuring points in $\mathcal{H}$ are distinguishable and unique with respect to $d$. In contrast, for a *pseudometric* $d$, $d\left(\boldsymbol{h}^{(1)}, \boldsymbol{h}^{(2)}\right) = 0$ does not necessarily imply that $\boldsymbol{h}^{(1)} = \boldsymbol{h}^{(2)}$, relaxing the distinguishability constraint of a metric. The following assumption is then imposed on the *pseudometric $d$*, leveraging results from kernel theory.

**Definition 2** (Conditionally positive definite kernel (Schölkopf, 2000)). *Let $\mathcal{H}$ be a non-empty set. A symmetric function $\tilde{k} : \mathcal{H} \times \mathcal{H} \to \mathbb{R}$ is a conditionally positive definite kernel on $\mathcal{H}$ if for all $N \in \mathbb{N}$ and $\boldsymbol{h}^{(1)}, \boldsymbol{h}^{(2)}, \ldots, \boldsymbol{h}^{(N)} \in \mathcal{H}$,*

$$\sum_{i=1}^{N} \sum_{j=1}^{N} c_i c_j \ \tilde{k}\left(\boldsymbol{h}^{(i)}, \boldsymbol{h}^{(j)}\right) \geq 0, \tag{4}$$

*with $c_1, c_2, \ldots, c_N \in \mathbb{R}$ and $\sum_{i=1}^{N} c_i = 0$.*

**Assumption 1.** *Let $d : \mathcal{H} \times \mathcal{H} \to \mathbb{R}_{\geq 0}$ be a pseudometric on $\mathcal{H}$ such that $-d^2$ is a conditionally positive definite kernel on $\mathcal{H}$.*

The Euclidean distance is an example of a *pseudometric* satisfying Assumption 1. A class of *pseudometrics* satisfying this assumption is provided below, see Schölkopf (2000) and Berg et al. (2012) for more.

**Remark 1.** *Consider the pseudometrics $d_1$ and $d_2$ on $\mathcal{H}$ satisfying Assumption 1. For $a > 0$ and $0 < p < 1$, the pseudometrics $a \cdot d_1$, $\sqrt{d_1^2 + d_2^2}$, and $d_1^p$ also satisfy Assumption 1.*

Assumption 1 thus offers considerable flexibility in the choice of *pseudometric $d$*. The following theorem then *softly* relaxes the injective and metric requirements in previous works.

**Theorem 1.** *Let $\mathcal{H}$ be a non-empty set with a pseudometric $d : \mathcal{H} \times \mathcal{H} \to \mathbb{R}_{\geq 0}$ satisfying Assumption 1. There exists a feature map $g : \mathcal{H} \to \mathcal{S}$ such that for every $\boldsymbol{h}^{(1)}, \boldsymbol{h}^{(2)} \in \mathcal{H}$ and $\varepsilon_1 > \varepsilon_2 > 0$,*

$$\varepsilon_2 < \left\| g\left(\boldsymbol{h}^{(1)}\right) - g\left(\boldsymbol{h}^{(2)}\right) \right\| < \varepsilon_1 \iff \varepsilon_2 < d\left(\boldsymbol{h}^{(1)}, \boldsymbol{h}^{(2)}\right) < \varepsilon_1. \tag{5}$$

Theorem 1 shows that, for each node $u \in \mathcal{V}$, given a *pseudometric* distance $d_u$ that represents a *dissimilarity* function on $\mathcal{H}$, possibly encoded with prior knowledge, there exists a corresponding feature map $g_u$ that maps distinct inputs $\boldsymbol{h_u}^{(1)}, \boldsymbol{h_u}^{(2)} \in \mathcal{H}$ close in the embedded feature space $\mathcal{S}$ if and only if $d_u$ determines $\boldsymbol{h_u}^{(1)}$ and $\boldsymbol{h_u}^{(2)}$ to be sufficiently *similar* on some representation. The lower bound $\varepsilon_2$ asserts the ability of $g_u$ to separate elements of $\mathcal{H}$ in the embedded feature space $\mathcal{S}$ while the upper bound $\varepsilon_1$ ensures that $g_u$ maintains the relationship between elements of $\mathcal{H}$ with respect to $d_u$. The feature map $g_u$ may then be described as *soft-injective*.[1] Fig. 1 provides an illustration of how the *soft-injective* feature map $g$ maps distinct elements of $\mathcal{H}$ to the same point in $\mathcal{S}$ since the corresponding *pseudometric* $d\left(\boldsymbol{h}^{(1)}, \boldsymbol{h}^{(2)}\right) = \left\| \left[\boldsymbol{h}^{(1)}\right]^2 - \left[\boldsymbol{h}^{(2)}\right]^2 \right\|$ deems these points to be *similar*. Corollary 1 extends this result for *multisets*.

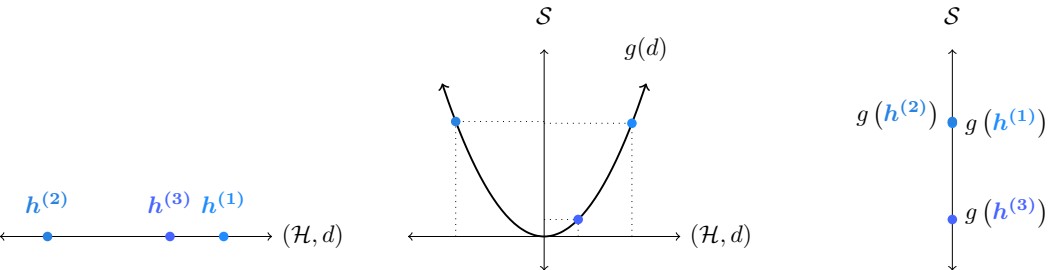

Figure 1: A *soft-injective* feature map $g : \mathcal{H} \to \mathcal{S}$ corresponding to a *pseudometric* $d$ on $\mathcal{H}$.

### 3.1 *Soft-isomorphic* relational graph convolution network

**Corollary 1.** *Let $\mathcal{H}$ be a non-empty set with a pseudometric $D$ on bounded, equinumerous multisets of $\mathcal{H}$ defined as*

$$D^2\left(\boldsymbol{H}^{(1)}, \boldsymbol{H}^{(2)}\right) := \sum_{\substack{\boldsymbol{h} \in \boldsymbol{H}^{(1)} \\ \boldsymbol{h}' \in \boldsymbol{H}^{(2)}}} d^2(\boldsymbol{h}, \boldsymbol{h}') - \frac{1}{2}\sum_{\substack{\boldsymbol{h} \in \boldsymbol{H}^{(1)} \\ \boldsymbol{h}' \in \boldsymbol{H}^{(1)}}} d^2(\boldsymbol{h}, \boldsymbol{h}') - \frac{1}{2}\sum_{\substack{\boldsymbol{h} \in \boldsymbol{H}^{(2)} \\ \boldsymbol{h}' \in \boldsymbol{H}^{(2)}}} d^2(\boldsymbol{h}, \boldsymbol{h}') \tag{6}$$

*for some pseudometric $d : \mathcal{H} \times \mathcal{H} \to \mathbb{R}_{\geq 0}$ satisfying Assumption 1 and bounded, equinumerous multisets $\boldsymbol{H}^{(1)}, \boldsymbol{H}^{(2)}$. There exists a feature map $g : \mathcal{H} \to \mathcal{S}$ such that for every $\boldsymbol{H}^{(1)}, \boldsymbol{H}^{(2)}$ and $\varepsilon_1 > \varepsilon_2 > 0$,*

$$\varepsilon_2 < \left\| G\left(\boldsymbol{H}^{(1)}\right) - G\left(\boldsymbol{H}^{(2)}\right) \right\| < \varepsilon_1 \iff \varepsilon_2 < D\left(\boldsymbol{H}^{(1)}, \boldsymbol{H}^{(2)}\right) < \varepsilon_1, \tag{7}$$

*where*

$$G(\boldsymbol{H}) = \sum_{\boldsymbol{h} \in \boldsymbol{H}} g(\boldsymbol{h}). \tag{8}$$

Similarly, Corollary 1 shows that, for each node $u \in \mathcal{V}$, given a *pseudometric* distance $D_u$ on *multisets* of $\mathcal{H}$ defined in Eqn. 6 with a corresponding *pseudometric* distance $d_u$ on $\mathcal{H}$, there exists a corresponding feature map $g_u$ and hash function $G_u$ defined in Eqn. 8 that produces *similar* outputs for distinct *multisets* of neighborhood features $\boldsymbol{H_u}^{(1)}, \boldsymbol{H_u}^{(2)}$ if and only if $D_u$ deems $\boldsymbol{H_u}^{(1)}$ and $\boldsymbol{H_u}^{(2)}$ to be sufficiently *similar* on some representation. Likewise, the lower and upper bounds guarantee the ability of $G_u$ to separate equinumerous *multisets* of $\mathcal{H}$ in the embedded feature space $\mathcal{S}$ while maintaining the relationship with respect to $D_u$. In this setting, the feature map $g_u$ may be interpreted as the *soft-injective* message function (Gilmer et al., 2017) that transforms the individual neighborhood features with a corresponding *soft-injective* hash function $G_u$. Meanwhile, the *pseudometric* $D_u$ corresponds to the kernel distance (Joshi et al., 2011) which intuitively represents the difference between the cross-distance and self-distance between two *multisets*. The two necessary properties of the *soft-injective* message function—*dynamic* and *anisotropic*—are then motivated below.

---

[1]The *pseudometric* $d$ induces the equivalence class $[\boldsymbol{h}]_d := \{\boldsymbol{h}' \in \mathcal{H} : d(\boldsymbol{h}, \boldsymbol{h}') = 0\}$ with the quotient space $\mathcal{H}_d := \mathcal{H} \setminus d = \left\{[\boldsymbol{h}]_d : \boldsymbol{h} \in \mathcal{H}\right\}$ such that $d$ becomes metric and the corresponding feature map $g$ becomes injective on $\mathcal{H}_d$ (Schoenberg, 1938).

***Dynamic* transformation**  To illustrate the role of the *pseudometric*, consider node $u$ with two neighbors $v_1$ and $v_2$ and the task of anomaly detection on the scalar node features $\boldsymbol{h_{v_1}}$ and $\boldsymbol{h_{v_2}}$ representing zero-mean scores. If $d_u$ simply corresponds to the Euclidean distance, then the corresponding hash function $G_u$ becomes linear as seen in Fig. 2a. Crucially, the contour plot highlights collisions—instances where distinct inputs produce identical outputs (*i.e.*, the equivalence class $[\boldsymbol{H}]_D$)—between *dissimilar multisets* of neighborhood features, resulting in aggregated neighborhood features that are less useful for the task.

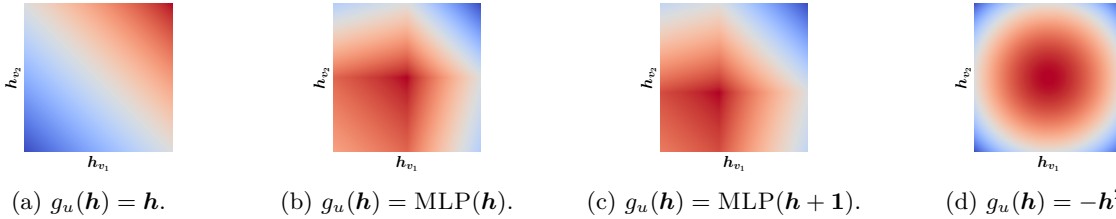

(a) $g_u(\boldsymbol{h}) = \boldsymbol{h}$.  (b) $g_u(\boldsymbol{h}) = \text{MLP}(\boldsymbol{h})$.  (c) $g_u(\boldsymbol{h}) = \text{MLP}(\boldsymbol{h} + \boldsymbol{1})$.  (d) $g_u(\boldsymbol{h}) = -\boldsymbol{h}^2$.

Figure 2: Hash functions $G_u$ under different message functions $g_u$.

Nevertheless, other choices of *pseudometrics*, possibly incorporating prior knowledge, would correspond to more complex message functions $g_u$. This leads to non-trivial hash functions $G_u$ and contour plots where only the regions determined by $D_u$ to be *similar* with respect to a given task may produce *similar* aggregated neighborhood features, making collisions more informative and controlled. In practice, this also highlights the significance of *dynamic* (Brody et al., 2022) (*i.e.*, a universal function approximator (Hornik et al., 1989)) message functions $g_u$ in the MPNN framework, which may be modeled as multi-layer perceptrons (MLPs) as illustrated in Figs. 2b and 2c.

As further illustration, if $d_u$ instead corresponds to the Euclidean distance of the squared score, then the corresponding hash function $G_u$ has the contour plot in Fig. 2d. The resulting hash collisions and equivalence classes then become more useful and meaningful for detecting anomalous scores.

***Anisotropic* messages**  It is also worth noting that Corollary 1 holds for each node $u \in \mathcal{V}$ independently. Hence, different nodes may correspond to different $D_u$, $d_u$, $g_u$, and $G_u$. For simplicity, especially in inductive learning contexts, consider a single *pseudometric* instead, defined as

$$D^2\left(\boldsymbol{H_u}^{(1)}, \boldsymbol{H_u}^{(2)}; \boldsymbol{h_u}\right) \coloneqq \sum_{\substack{\boldsymbol{h} \in \boldsymbol{H_u}^{(1)} \\ \boldsymbol{h'} \in \boldsymbol{H_u}^{(2)}}} d^2(\boldsymbol{h}, \boldsymbol{h'}; \boldsymbol{h_u}) - \frac{1}{2}\sum_{\substack{\boldsymbol{h} \in \boldsymbol{H_u}^{(1)} \\ \boldsymbol{h'} \in \boldsymbol{H_u}^{(1)}}} d^2(\boldsymbol{h}, \boldsymbol{h'}; \boldsymbol{h_u}) - \frac{1}{2}\sum_{\substack{\boldsymbol{h} \in \boldsymbol{H_u}^{(2)} \\ \boldsymbol{h'} \in \boldsymbol{H_u}^{(2)}}} d^2(\boldsymbol{h}, \boldsymbol{h'}; \boldsymbol{h_u}), \quad (9)$$

with a single hash function, defined as

$$G\left(\boldsymbol{H_u}; \boldsymbol{h_u}\right) \coloneqq \sum_{\boldsymbol{h} \in \boldsymbol{H_u}} g\left(\boldsymbol{h}; \boldsymbol{h_u}\right), \tag{10}$$

for every node $u \in \mathcal{V}$. This approach makes $D$, $d$, $g$, and $G$ *anisotropic* (Dwivedi et al., 2023) (*i.e.*, a function of both the features of the query (center) node $\boldsymbol{h_u}$ and key (neighboring) nodes $\boldsymbol{h} \in \boldsymbol{H_u}$). In addition, contextualized on the features of the query node, $D$ may still be interpreted as a *pseudometric* controlling hash collisions with a corresponding *soft-injective* hash function $G$.

Furthermore, the integration of $\boldsymbol{h_u}$ also allows for the interpretation of $g$ as a *soft-injective* relational message function, guiding how features of the key nodes are to be embedded and transformed based on the features of the query node. Figs. 2b and 2c provide intuition for this idea where the introduction of a bias term, assuming a function of the features of the query node, shifts the contour plot to produce distinct aggregated neighborhood features $\boldsymbol{a_u} \neq \boldsymbol{a_{u'}}$ for nodes $u \neq u' \in \mathcal{V}$ with identical neighborhood features $\boldsymbol{H_u} = \boldsymbol{H_{u'}}$ but distinct features $\boldsymbol{h_u} \neq \boldsymbol{h_{u'}}$. Moreover, one may also inject stochasticity into the node features to distinguish between nodes $u \neq u' \in \mathcal{V}$ with identical features $\boldsymbol{h_u} = \boldsymbol{h_{u'}}$ and neighborhood features $\boldsymbol{H_u} = \boldsymbol{H_{u'}}$ with high probability (Sato et al., 2021) and to imitate having distinct $D_u$, $d_u$, $g_u$, and $G_u$ for each node $u \in \mathcal{V}$.

**Proposed model** For a graph representation learning problem, one may directly model the *anisotropic* and *dynamic soft-injective* relational message function $g$ as a two-layer MLP, with implicitly learned *pseudometrics*, to obtain the *soft-isomorphic* relational graph convolution network (SIR-GCN)

$$h_u^* = \sum_{v \in \mathcal{N}(u)} W_R \, \sigma \left( W_Q h_u + W_K h_v \right), \tag{11}$$

where $\sigma$ is a non-linear activation function, $W_Q, W_K \in \mathbb{R}^{d_{\text{hidden}} \times d_{\text{in}}}$, and $W_R \in \mathbb{R}^{d_{\text{out}} \times d_{\text{hidden}}}$. Leveraging linearity, the model has a computational complexity of

$$\mathcal{O} \left( |\mathcal{V}| \times d_{\text{hidden}} \times d_{\text{in}} + |\mathcal{E}| \times d_{\text{hidden}} + |\mathcal{V}| \times d_{\text{out}} \times d_{\text{hidden}} \right) \tag{12}$$

with only the application of an activation function along edges, making it comparable to classical GNNs in literature. Nevertheless, $\sigma$ may also be modeled as a deep MLP in practice if modeling $g$ as a shallow two-layer MLP becomes infeasible.

In essence, SIR-GCN is a simple instance of the MPNN framework explicitly designed to handle uncountable node features while maintaining rigorous theoretical foundations. It emphasizes the *anisotropic* and *dynamic* transformation of neighborhood features, obtaining contextualized messages that enable it to learn complex relationships between neighboring nodes. Moreover, the proposed model is also computationally efficient, requiring only a single aggregator and applying only an activation function along edges to facilitate effective message-passing of uncountable node features.

### 3.2 *Soft-isomorphic* graph readout function

Corollary 1 also shows that, for each graph $\mathcal{G}$, given a *pseudometric* distance $D_{\mathcal{G}}$ on *multisets* of $\mathcal{H}$ defined in Eqn. 6 with a corresponding *pseudometric* distance $d_{\mathcal{G}}$ on $\mathcal{H}$, there exists a corresponding feature map $r_{\mathcal{G}}$ and graph readout function $R_{\mathcal{G}}$ defined in Eqn. 8. While this result holds for each graph $\mathcal{G}$ independently, one may simply consider a single $D$, $d$, $r$, and $R$ for every graph $\{\mathcal{G}_d\}_{d \in \mathcal{D}}$ under task $\mathcal{D}$. Nevertheless, the graph context and structure may also be integrated into $D$, $d$, $r$, and $R$, through a virtual super node (Gilmer et al., 2017) for instance, to imitate having distinct $D_{\mathcal{G}}$, $d_{\mathcal{G}}$, $r_{\mathcal{G}}$, and $R_{\mathcal{G}}$ for each graph $\mathcal{G}$ and to further enhance its representational capability.

Similarly, for a graph representation learning problem, the *dynamic soft-injective* feature map $r$ may also be directly modeled as an MLP, with implicitly learned *pseudometrics*, to obtain the *soft-isomorphic* graph readout function

$$h_{\mathcal{G}} = \sum_{v \in \mathcal{V}_{\mathcal{G}}} \text{MLP}_R \left( h_v \right), \tag{13}$$

where $\text{MLP}_R$ corresponds to $r$ and $h_{\mathcal{G}}$ is the graph-level feature of graph $\mathcal{G}$.

## 4 Mathematical discussion

The mathematical relationship between SIR-GCN and key GNNs in literature—GCN, GraphSAGE, GAT, GIN, and PNA—are presented in this section to underscore the contribution and distinctive advantages of the proposed model. It is worth noting that while activation functions and MLPs applied after each GNN layer play a significant role in the overall performance, the discussions only focus on the core message-passing operation that defines GNNs. In addition, the relationship between SIR-GCN and the 1-WL test is also presented to contextualize the representational capability of the former.

### 4.1 GCN and GraphSAGE

It may be shown that Corollary 1 holds up to a constant scale. Hence, the mean aggregation and symmetric mean aggregation, by extension, may be used in place of the sum aggregation in Eqn. 11. If one sets $\sigma$ as identity or $\text{PReLU}(\alpha = 1)$, $W_Q = 0$, $W_R W_K = W$, and $\tilde{\mathcal{N}}(u) := \mathcal{N}(u) \cup \{u\}$, one obtains

$$h_u^* = \sum_{v \in \mathcal{N}(u)} \frac{1}{\sqrt{|\mathcal{N}(u)|}\sqrt{|\mathcal{N}(v)|}} W h_v \tag{14}$$

and

$$h_u^* = \frac{1}{\left|\tilde{\mathcal{N}}(u)\right|} \sum_{v \in \tilde{\mathcal{N}}(u)} W h_v \tag{15}$$

which are equivalent to GCN and GraphSAGE with mean aggregation, respectively. Moreover, the sum aggregation may also be replaced with the max aggregation, albeit without theoretical justification, to recover GraphSAGE with max pooling. Thus, GCN and GraphSAGE[2] may be viewed as instances of SIR-GCN. The key difference lies in the *isotropic* (Dwivedi et al., 2023) nature (*i.e.*, a function of only the features of the key nodes) of GCN and GraphSAGE and their non-linearities only after aggregating the neighborhood features.

## 4.2 GAT

Moreover, in Brody et al. (2022), the attention mechanism of GATv2 is modeled as an MLP given by

$$e_{u,v} = \boldsymbol{a}_{\text{GAT}}^\top \, \text{LEAKYRELU} \left( \boldsymbol{W}_{Q,\text{GAT}} \, \boldsymbol{h_u} + \boldsymbol{W}_{K,\text{GAT}} \, \boldsymbol{h_v} \right), \tag{16}$$

with the message from node $v$ to node $u$ proportional to $\exp(e_{u,v}) \cdot \boldsymbol{W}_{K,\text{GAT}} \, \boldsymbol{h_v}$. While the attention mechanism of GATv2 is *anisotropic* and *dynamic*, its messages are nevertheless only linearly transformed with the query node $u$ only determining the degree of contribution of each message through the scalar $e_{u,v}$. Meanwhile, SIR-GCN directly applies the *anisotropic* and *dynamic* function in Eqn. 16 to the message function, allowing the features of the query node to *dynamically* transform messages. Specifically, if $\boldsymbol{W_Q} = \boldsymbol{W}_{Q,\text{GAT}}$, $\boldsymbol{W_K} = \boldsymbol{W}_{K,\text{GAT}}$, $\sigma = \text{LEAKYRELU}$ and $\boldsymbol{W_R} = \boldsymbol{a}_{\text{GAT}}^\top$, one obtains

$$h_u^* = \sum_{v \in \mathcal{N}(u)} \boldsymbol{a}_{\text{GAT}}^\top \, \text{LEAKYRELU} \left( \boldsymbol{W}_{Q,\text{GAT}} \, \boldsymbol{h_u} + \boldsymbol{W}_{K,\text{GAT}} \, \boldsymbol{h_v} \right) \tag{17}$$

which shows Eqn. 16 becoming a contextualized message in SIR-GCN. Nevertheless, GAT and GATv2 may also be recovered, up to a normalizing constant, with the appropriate parameters.

## 4.3 GIN

Likewise, within the proposed SIR-GCN, one may explicitly add a residual connection in the combination strategy to obtain

$$h_u^* = \text{MLP}_{\text{Res}}(\boldsymbol{h_u}) + \sum_{v \in \mathcal{N}(u)} \boldsymbol{W_R} \, \sigma \left( \boldsymbol{W_Q} \boldsymbol{h_u} + \boldsymbol{W_K} \boldsymbol{h_v} \right), \tag{18}$$

where $\text{MLP}_{\text{Res}}$ is a learnable residual network. If $\text{MLP}_{\text{Res}}(\boldsymbol{h}) = (1 + \epsilon) \cdot \boldsymbol{h}$, $\sigma = \text{PRELU}(\alpha = 1)$, $\boldsymbol{W_Q} = \boldsymbol{0}$, and $\boldsymbol{W_R} \boldsymbol{W_K} = \boldsymbol{I}$, then

$$h_u^* = (1 + \epsilon) \cdot \boldsymbol{h_u} + \sum_{v \in \mathcal{N}(u)} \boldsymbol{h_v} \tag{19}$$

is equivalent to GIN. Hence, SIR-GCN with residual connection generalizes GIN.

## 4.4 PNA

Furthermore, while SIR-GCN and PNA approach the problem of uncountable node features differently, both models highlight the significance of *anisotropic* message functions considering both the features of the query and key nodes. The key difference lies with PNA using multiple aggregators and a *static* (Brody et al., 2022) (*i.e.*, a function approximator with limited expressivity; *e.g.*, linear) message function

$$m \left( \boldsymbol{h_v}, \boldsymbol{h_u} \right) = \boldsymbol{W_K} \boldsymbol{h_v} + \boldsymbol{W_Q} \boldsymbol{h_u} =: \boldsymbol{W_K} \boldsymbol{h_v} + \boldsymbol{b_u}. \tag{20}$$

Consequently, the influence of the query node on the aggregated neighborhood feature becomes limited. In particular, when using mean, max, or min aggregators, the influence of the query node $u$ is restricted to the bias term $\boldsymbol{b_u} := \boldsymbol{W_Q} \boldsymbol{h_u}$. Moreover, with normalized moment aggregators, the bias term is effectively canceled

---

[2]GraphSAGE with LSTM aggregation is not included in this discussion.

out during the normalization process, further reducing the influence of the query node. Hence, PNA does not fully leverage its *anisotropic* nature, attributed to its heuristic application of multiple aggregators and scalers in a linear MPNN, thereby limiting its expressivity. In contrast, the *dynamic* nature of SIR-GCN allows for the non-linear and contextualized embedding of the features of the query node within the messages, thereby fully leveraging its *anisotropic* nature while allowing it to employ only a single aggregator.

### 4.5 1-WL test

Additionally, in terms of graph isomorphism representational capability, SIR-GCN is comparable to a modified 1-WL test. Suppose $w_u^{(l)}$ is the WL node label of node $u$ at the $l$th WL-test iteration. The modified update equation is given by

$$w_u^{(l)} \leftarrow \text{hash}\left(\left\{\!\!\left\{ \left[w_v^{(l-1)}, w_u^{(l-1)}\right] : v \in \mathcal{N}(u) \right\}\!\!\right\}\right), \tag{21}$$

where the modification lies in concatenating the label of the query node $u$ with every element of the *multiset* before hashing. This modification, while negligible when $\mathcal{H}$ is countable, becomes significant when $\mathcal{H}$ is uncountable as highlighted in the previous section. Thus, SIR-GCN inherits the theoretical capabilities of the 1-WL test.

### 4.6 SIR-GCN

Overall, SIR-GCN is a simple MPNN instance that offers flexibility in two key dimensions of GNN—message transformation and aggregator. Consequently, it generalizes four classical GNNs in literature—GCN, GraphSAGE, GAT, and GIN—ensuring that it is at least as expressive as these models. Notably, SIR-GCN distinguishes itself from other GNNs by employing both *anisotropic* and *dynamic* (*i.e.*, contextualized) messages within the MPNN framework, enabling the non-uniform aggregation of neighboring nodes in heterophilous graphs (Zheng et al., 2024) while maintaining adaptability to homophilous graphs.

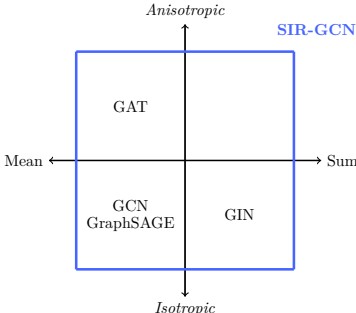

Figure 3: SIR-GCN generalizes classical GNNs.

In addition, SIR-GCN also distinguishes itself from PNA in addressing the problem of uncountable node features by employing only a single aggregator that theoretically holds for graphs of arbitrary sizes, thus reducing computational complexity. Nevertheless, its expressivity is maintained through contextualized messages via the application of only an activation function along edges, allowing it to inherit the representational capability of the 1-WL test.

## 5 Experiments

Experiments on synthetic and benchmark datasets in node and graph property prediction tasks are performed to highlight the expressivity of SIR-GCN. Following the evaluation methodology of Xu et al. (2019), Corso et al. (2020), and Brody et al. (2022), the key GNNs in literature without advanced architectural design nor domain-specific features are used as primary comparisons to ensure a fair evaluation.

### 5.1 Synthetic datasets

**DictionaryLookup** DictionaryLookup (Brody et al., 2022) consists of bipartite graphs with $2n$ nodes—$n$ *key* nodes each with an attribute and value and $n$ *query* nodes each with an attribute. The task is to predict the values of *query* nodes by matching their attributes with the *key* nodes as seen in Fig. 4.

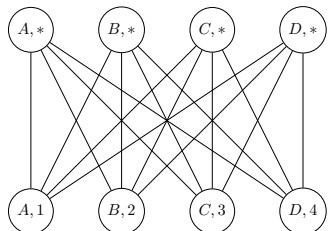

Figure 4: DictionaryLookup.

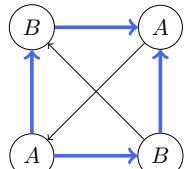

Figure 5: HeteroEdgeCount.

Table 1: Test accuracy on DictionaryLookup.

| Model | $n = 10$ | $n = 20$ | $n = 30$ | $n = 40$ | $n = 50$ |
|---|---|---|---|---|---|
| GCN | $0.10 \pm 0.00$ | $0.05 \pm 0.00$ | $0.03 \pm 0.00$ | $0.03 \pm 0.00$ | $0.02 \pm 0.00$ |
| GraphSAGE | $0.10 \pm 0.00$ | $0.05 \pm 0.00$ | $0.03 \pm 0.00$ | $0.02 \pm 0.00$ | $0.02 \pm 0.00$ |
| GATv2 | $0.99 \pm 0.03$ | $0.88 \pm 0.18$ | $0.74 \pm 0.28$ | $0.56 \pm 0.37$ | $0.60 \pm 0.40$ |
| GIN | $0.78 \pm 0.07$ | $0.29 \pm 0.03$ | $0.12 \pm 0.03$ | $0.03 \pm 0.00$ | $0.02 \pm 0.01$ |
| PNA | $1.00 \pm 0.00$ | $0.97 \pm 0.02$ | $0.86 \pm 0.09$ | $0.66 \pm 0.09$ | $0.50 \pm 0.05$ |
| SIR-GCN | $\mathbf{1.00 \pm 0.00}$ | $\mathbf{1.00 \pm 0.00}$ | $\mathbf{1.00 \pm 0.00}$ | $\mathbf{1.00 \pm 0.00}$ | $\mathbf{1.00 \pm 0.00}$ |

Table 1 presents the mean and standard deviation of the test accuracy for SIR-GCN, GCN, GraphSAGE, GATv2, GIN, and PNA across different values of $n$. Notably, SIR-GCN and GATv2 can achieve perfect accuracy in this synthetic task attributed to their *anisotropic* and *dynamic* nature, enabling them to learn the relationship between every *query* and *key* node. Nonetheless, it may be observed that GATv2 suffers from performance degradation in some trials. Meanwhile, the other models fail to predict the value of *query* nodes even for the training graphs due to their *isotropic* and/or *static* nature, hindering their ability to learn relationships between neighboring nodes. The results hence underscore the utility of a *dynamic* attentional or relational mechanism in capturing the relationship between the *query* and *key* nodes.

**HeteroEdgeCount**  HeteroEdgeCount is an original synthetic dataset consisting of randomly generated directed graphs with each node randomly labeled one of $c$ classes. The task is then to count the number of heterophilous directed edges in each graph connecting nodes with different class labels as illustrated in Fig. 5. This dataset is explicitly designed to highlight the limitations of key GNNs in literature, particularly the theoretically grounded GNNs such as GIN and PNA, even in trivial tasks involving countable node features. Specifically, it underscores the utility of *anisotropic* and *dynamic* message functions in learning the relationships between neighboring nodes. Crucially, this dataset is not intended to assess the ability of GNNs to handle heterophilous graphs, as this falls beyond the scope of this work.

Table 2: Test mean squared error on HeteroEdgeCount.

| Model | $c = 2$ | $c = 4$ | $c = 6$ | $c = 8$ | $c = 10$ |
|---|---|---|---|---|---|
| GCN | $22749 \pm 1242$ | $50807 \pm 2828$ | $62633 \pm 3491$ | $68965 \pm 3784$ | $72986 \pm 4025$ |
| GraphSAGE | $22962 \pm 1215$ | $36854 \pm 2330$ | $30552 \pm 1574$ | $21886 \pm 1896$ | $16529 \pm 1589$ |
| GATv2 | $22329 \pm 1307$ | $44972 \pm 2834$ | $49940 \pm 2942$ | $50063 \pm 3407$ | $49661 \pm 3488$ |
| GIN | $39.620 \pm 2.060$ | $37.193 \pm 1.382$ | $34.649 \pm 1.502$ | $32.424 \pm 1.841$ | $30.091 \pm 1.429$ |
| PNA | $172.15 \pm 97.82$ | $224.83 \pm 85.80$ | $249.99 \pm 108.56$ | $251.49 \pm 98.84$ | $195.72 \pm 36.65$ |
| SIR-GCN | $\mathbf{0.001 \pm 0.000}$ | $\mathbf{0.004 \pm 0.005}$ | $\mathbf{1.495 \pm 4.428}$ | $\mathbf{0.038 \pm 0.068}$ | $\mathbf{0.089 \pm 0.134}$ |

Table 2 presents the mean and standard deviation of the test mean squared error (MSE) for SIR-GCN, GCN, GraphSAGE, GATv2, GIN, and PNA across different values of $c$. Notably, SIR-GCN consistently achieves near-zero MSE loss due to its *anisotropic* and *dynamic* nature as well as its sum aggregation, allowing it to learn the relationship between the labels of neighboring nodes while retaining the graph structure. In fact, for $\boldsymbol{W_Q} = \boldsymbol{I}$, $\boldsymbol{W_K} = -\boldsymbol{I}$, $\sigma = \mathrm{ReLU}$, and $\boldsymbol{W_R} = \mathbf{1}^\top$, it may be shown that SIR-GCN will always produce the correct output for every graph. In contrast, GCN, GraphSAGE, and GATv2 obtained large MSE losses due to their mean or max aggregation which fails to preserve the graph structure as noted by Xu et al. (2019).

Meanwhile, GIN and PNA successfully retain the graph structure with their sum aggregation but fail to differentiate neighboring nodes due to their *static* nature. The results thus illustrate the utility of *anisotropic* and *dynamic* message functions using sum aggregation even in simple tasks with countable node features, highlighting the limitations of existing GNNs.

## 5.2 Benchmark datasets

**Benchmarking GNNs**  Benchmarking GNNs (Dwivedi et al., 2023) is a collection of benchmark datasets consisting of diverse mathematical and real-world graphs across various GNN tasks. In particular, the WikiCS, PATTERN, and CLUSTER datasets fall under node property prediction tasks while the MNIST, CIFAR10, and ZINC datasets fall under graph property prediction tasks. Furthermore, the WikiCS, MNIST, and CIFAR10 datasets have uncountable node features while the remaining datasets have countable node features. The performance metric of ZINC is the mean absolute error (MAE) while the performance metric of the remaining datasets is accuracy. These six benchmark datasets encompass a diverse range of GNN tasks, enabling a comprehensive and robust evaluation of model performance. Dwivedi et al. (2023) provides more information regarding the individual datasets.

Table 3: Test performance on Benchmarking GNNs.

| Model | WikiCS (↑) | PATTERN (↑) | CLUSTER (↑) | MNIST (↑) | CIFAR10 (↑) | ZINC (↓) |
|---|---|---|---|---|---|---|
| GCN | $77.47 \pm 0.85$ | $85.50 \pm 0.05$ | $47.83 \pm 1.51$ | $90.12 \pm 0.15$ | $54.14 \pm 0.39$ | $0.416 \pm 0.006$ |
| GraphSAGE | $74.77 \pm 0.95$ | $50.52 \pm 0.00$ | $50.45 \pm 0.15$ | $97.31 \pm 0.10$ | $65.77 \pm 0.31$ | $0.468 \pm 0.003$ |
| GAT | $76.91 \pm 0.82$ | $75.82 \pm 1.82$ | $57.73 \pm 0.32$ | $95.54 \pm 0.21$ | $64.22 \pm 0.46$ | $0.475 \pm 0.007$ |
| GATv2 | - | - | - | - | $67.48 \pm 0.53$ | $0.447 \pm 0.015$ |
| GIN | $75.86 \pm 0.58$ | $85.59 \pm 0.01$ | $58.38 \pm 0.24$ | $96.49 \pm 0.25$ | $55.26 \pm 1.53$ | $0.387 \pm 0.015$ |
| PNA | - | - | - | $97.19 \pm 0.08$ | $70.21 \pm 0.15$ | $0.320 \pm 0.032$ |
| EGC-S | - | - | - | - | $66.92 \pm 0.37$ | $0.364 \pm 0.020$ |
| EGC-M | - | - | - | - | $71.03 \pm 0.42$ | $0.281 \pm 0.007$ |
| SIR-GCN | $\mathbf{78.06 \pm 0.66}$ | $\mathbf{85.75 \pm 0.03}$ | $\mathbf{63.35 \pm 0.19}$ | $\mathbf{97.90 \pm 0.08}$ | $\mathbf{71.98 \pm 0.40}$ | $\mathbf{0.278 \pm 0.024}$ |

Note: Missing values indicate that no results were published in previous works.

Table 3 presents the mean and standard deviation of the test performance for SIR-GCN, GCN, GraphSAGE, GAT, GATv2, GIN, and PNA across the six benchmark datasets with the experimental set-up, such as parameter count and model architecture, following that of Dwivedi et al. (2023), Corso et al. (2020), and Tailor et al. (2022) to ensure a fair evaluation where performance differences are solely attributed to the GNN architectural design. The test performance for the efficient graph convolution single (EGC-S) and efficient graph convolution multiple (EGC-M) (Tailor et al., 2022) are also presented as additional MPNN-based baselines. Notably, SIR-GCN consistently outperforms classical GNNs in literature by a substantial margin which may be attributed to its ability to generalize these models, complementing the mathematical discussions in the previous section. Moreover, despite employing multiple aggregators and incurring higher computational complexity to ensure injectivity, PNA still fails to outperform the simpler and more computationally efficient SIR-GCN on datasets with uncountable node features, even though the former is explicitly designed for such tasks. Furthermore, SIR-GCN also outperforms the more recent EGC-S and EGC-M despite their use of additional tricks, including multiple convolutional basis weights, regularization heads, and aggregators. Overall, the results underscore that, under the same constraints, SIR-GCN consistently outperforms MPNN-based baselines despite its simplicity, establishing it as a promising alternative to existing MPNNs.

**Open Graph Benchmark**  Open Graph Benchmark (Hu et al., 2020a) is another collection of datasets consisting of realistic, large-scale, and diverse benchmarks for GNNs. In particular, the ogbn-arxiv with its uncountable node features falls under node property prediction tasks. Meanwhile, the ogbg-molhiv with its countable node features falls under graph property prediction tasks. The performance metric of ogbn-arxiv is accuracy while the performance metric of ogbg-molhiv is the area under the receiver operating characteristic curve (ROC-AUC). Hu et al. (2020a) provides more information regarding the individual datasets.

Table 4: Test performance on Open Graph Benchmark.

| Model | ogbn-arxiv ($\uparrow$) | ogbg-molhiv ($\uparrow$) |
|---|---|---|
| GCN | $71.92 \pm 0.21$ | $76.14 \pm 1.29$ |
| GraphSAGE | $71.73 \pm 0.26$ | $75.97 \pm 1.69$ |
| GAT | $71.81 \pm 0.23$ | $77.17 \pm 1.37$ |
| GATv2 | $71.87 \pm 0.43$ | $77.15 \pm 1.55$ |
| GIN | $67.33 \pm 1.47$ | $76.02 \pm 1.35$ |
| PNA | $71.21 \pm 0.30$ | $\mathbf{79.05 \pm 1.32}$ |
| EGC-S | $72.21 \pm 0.17$ | $77.44 \pm 1.08$ |
| EGC-M | $71.96 \pm 0.23$ | $78.18 \pm 1.53$ |
| SIR-GCN | $\mathbf{72.52 \pm 0.16}$ | $77.63 \pm 0.84$ |

Table 4 presents the mean and standard deviation of the test performance for SIR-GCN, GCN, GraphSAGE, GAT, GATv2, GIN, PNA, EGC-S, and EGC-M across the two large-scale benchmark datasets with the experimental set-up following that of Corso et al. (2020) and Tailor et al. (2022) to ensure a fair evaluation. Notably, SIR-GCN still outperforms both classical GNNs and EGC-S by a significant margin even in these large-scale graphs, which may be attributed to its *anisotropic* and *dynamic* message function. Unsurprisingly, however, PNA and EGC-M exhibit better performance relative to SIR-GCN on ogbg-molhiv as this molecular property prediction task greatly benefits from maintaining graph isomorphism via multiple aggregators, scalers, convolutional basis weights, and regularization heads. Nevertheless, SIR-GCN outperforms both PNA and EGC-M by a substantial margin on ogbn-arxiv as this 40-class node classification task does not require maintaining graph isomorphism. Overall, the results highlight the significance of contextualized messages in enhancing GNN expressivity and the utility of *softly* relaxing the injective and metric requirements within the MPNN framework for *most* practical GNN applications.

## 6 Conclusion

In summary, the paper provides a new perspective for creating powerful GNNs when the space of node features is uncountable. The key idea is to use *pseudometric* distances on the space of input to create *soft-injective* functions such that distinct inputs may produce *similar* outputs if and only if the distance between the inputs is sufficiently small on some representation. From the theoretical results, SIR-GCN is proposed as a simple and computationally efficient MPNN instance emphasizing contextualized message transformation. Notably, compared to existing MPNN instances, this distinctive feature enables it to learn complex relationships between neighboring nodes and allows it to better handle uncountable node features. Furthermore, the proposed model is also demonstrated to generalize classical GNN methodologies. Despite its simple architectural design and minimal computational requirements, empirical results on synthetic and benchmark datasets underscore the expressivity of SIR-GCN, making it a promising candidate for practical GNN applications. Overall, the paper contributes to GNN literature by theoretically and empirically demonstrating the necessity of both *anisotropic* and *dynamic* messages to enhance GNN expressivity. Future works may extend the present framework by considering more complex *pseudometric* formulations for bounded, equinumerous *multisets* of $\mathcal{H}$ in Corollary 1. They may also consider a formal analysis of the relationship between contextualized messages and performance on heterophilous graph tasks.

### Acknowledgments

This work is supported in part by the Japan Society for the Promotion of Science through the Grants-in-Aid for Scientific Research Program (KAKENHI 18K19821) and in part by Kyoto University and Toyota Motor Corporation through the joint project titled "Advanced Mathematical Science for Mobility Society."

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

## A  Proofs

**Theorem 2** (Hilbert space representation of conditionally positive definite kernels (Schoenberg, 1938; Schölkopf, 2000; Berg et al., 2012))**.** *Let $\mathcal{H}$ be a non-empty set and $\tilde{k} : \mathcal{H} \times \mathcal{H} \to \mathbb{R}$ a conditionally positive definite kernel on $\mathcal{H}$ satisfying $\tilde{k}(\boldsymbol{h}, \boldsymbol{h}) = 0$ for all $\boldsymbol{h} \in \mathcal{H}$. There exists a Hilbert space $\mathcal{S}$ of real-valued functions on $\mathcal{H}$ and a feature map $g : \mathcal{H} \to \mathcal{S}$ such that for every $\boldsymbol{h}^{(1)}, \boldsymbol{h}^{(1)} \in \mathcal{H}$,*

$$\left\| g\left(\boldsymbol{h}^{(1)}\right) - g\left(\boldsymbol{h}^{(2)}\right) \right\|^2 = -\tilde{k}\left(\boldsymbol{h}^{(1)}, \boldsymbol{h}^{(2)}\right). \tag{22}$$

*Proof.* See Schölkopf (2000). $\qquad\square$

**Theorem 1.** *Let $\mathcal{H}$ be a non-empty set with a pseudometric $d : \mathcal{H} \times \mathcal{H} \to \mathbb{R}_{\geq 0}$ satisfying Assumption 1. There exists a feature map $g : \mathcal{H} \to \mathcal{S}$ such that for every $\boldsymbol{h}^{(1)}, \boldsymbol{h}^{(2)} \in \mathcal{H}$ and $\varepsilon_1 > \varepsilon_2 > 0$,*

$$\varepsilon_2 < \left\| g\left(\boldsymbol{h}^{(1)}\right) - g\left(\boldsymbol{h}^{(2)}\right) \right\| < \varepsilon_1 \iff \varepsilon_2 < d\left(\boldsymbol{h}^{(1)}, \boldsymbol{h}^{(2)}\right) < \varepsilon_1. \tag{5}$$

*Proof.* Let $d : \mathcal{H} \times \mathcal{H} \to \mathbb{R}_{\geq 0}$ be a *pseudometric* satisfying Assumption 1. By Theorem 2, there exists a feature map $g : \mathcal{H} \to \mathcal{S}$ such that for every $\boldsymbol{h}^{(1)}, \boldsymbol{h}^{(2)} \in \mathcal{H}$,

$$\left\| g\left(\boldsymbol{h}^{(1)}\right) - g\left(\boldsymbol{h}^{(2)}\right) \right\| = d\left(\boldsymbol{h}^{(1)}, \boldsymbol{h}^{(2)}\right). \tag{23}$$

Hence, for every $\varepsilon_1 > \varepsilon_2 > 0$,

$$\varepsilon_2 < \left\| g\left(\boldsymbol{h}^{(1)}\right) - g\left(\boldsymbol{h}^{(2)}\right) \right\| < \varepsilon_1 \iff \varepsilon_2 < d\left(\boldsymbol{h}^{(1)}, \boldsymbol{h}^{(2)}\right) < \varepsilon_1. \tag{24}$$

$\qquad\square$

**Theorem 3.** *Suppose $\boldsymbol{h}^{(0)}, \boldsymbol{h}^{(1)}, \boldsymbol{h}^{(2)} \in \mathcal{H}$ and $\tilde{k} : \mathcal{H} \times \mathcal{H} \to \mathbb{R}$ is a symmetric function. Then*

$$k\left(\boldsymbol{h}^{(1)}, \boldsymbol{h}^{(2)}\right) \coloneqq \frac{1}{2}\left[\tilde{k}\left(\boldsymbol{h}^{(1)}, \boldsymbol{h}^{(2)}\right) - \tilde{k}\left(\boldsymbol{h}^{(1)}, \boldsymbol{h}^{(0)}\right) - \tilde{k}\left(\boldsymbol{h}^{(0)}, \boldsymbol{h}^{(2)}\right) + \tilde{k}\left(\boldsymbol{h}^{(0)}, \boldsymbol{h}^{(0)}\right)\right] \tag{25}$$

*is positive definite if and only if $\tilde{k}$ is conditionally positive definite.*

*Proof.* See Schölkopf (2000). $\qquad\square$

**Corollary 1.** *Let $\mathcal{H}$ be a non-empty set with a pseudometric $D$ on bounded, equinumerous multisets of $\mathcal{H}$ defined as*

$$D^2\left(\boldsymbol{H}^{(1)}, \boldsymbol{H}^{(2)}\right) \coloneqq \sum_{\substack{\boldsymbol{h} \in \boldsymbol{H}^{(1)} \\ \boldsymbol{h}' \in \boldsymbol{H}^{(2)}}} d^2(\boldsymbol{h}, \boldsymbol{h}') - \frac{1}{2}\sum_{\substack{\boldsymbol{h} \in \boldsymbol{H}^{(1)} \\ \boldsymbol{h}' \in \boldsymbol{H}^{(1)}}} d^2(\boldsymbol{h}, \boldsymbol{h}') - \frac{1}{2}\sum_{\substack{\boldsymbol{h} \in \boldsymbol{H}^{(2)} \\ \boldsymbol{h}' \in \boldsymbol{H}^{(2)}}} d^2(\boldsymbol{h}, \boldsymbol{h}') \tag{6}$$

*for some pseudometric $d : \mathcal{H} \times \mathcal{H} \to \mathbb{R}_{\geq 0}$ satisfying Assumption 1 and bounded, equinumerous multisets $\boldsymbol{H}^{(1)}, \boldsymbol{H}^{(2)}$. There exists a feature map $g : \mathcal{H} \to \mathcal{S}$ such that for every $\boldsymbol{H}^{(1)}, \boldsymbol{H}^{(2)}$ and $\varepsilon_1 > \varepsilon_2 > 0$,*

$$\varepsilon_2 < \left\| G\left(\boldsymbol{H}^{(1)}\right) - G\left(\boldsymbol{H}^{(2)}\right) \right\| < \varepsilon_1 \iff \varepsilon_2 < D\left(\boldsymbol{H}^{(1)}, \boldsymbol{H}^{(2)}\right) < \varepsilon_1, \tag{7}$$

*where*

$$G(\boldsymbol{H}) = \sum_{\boldsymbol{h} \in \boldsymbol{H}} g(\boldsymbol{h}). \tag{8}$$

*Proof.* Let $D$ be a *pseudometric* on bounded, equinumerous *multisets* of $\mathcal{H}$ defined as

$$D^2\left(\boldsymbol{H}^{(1)}, \boldsymbol{H}^{(2)}\right) \coloneqq \sum_{\substack{\boldsymbol{h} \in \boldsymbol{H}^{(1)} \\ \boldsymbol{h}' \in \boldsymbol{H}^{(2)}}} d^2(\boldsymbol{h}, \boldsymbol{h}') - \frac{1}{2} \sum_{\substack{\boldsymbol{h} \in \boldsymbol{H}^{(1)} \\ \boldsymbol{h}' \in \boldsymbol{H}^{(1)}}} d^2(\boldsymbol{h}, \boldsymbol{h}') - \frac{1}{2} \sum_{\substack{\boldsymbol{h} \in \boldsymbol{H}^{(2)} \\ \boldsymbol{h}' \in \boldsymbol{H}^{(2)}}} d^2(\boldsymbol{h}, \boldsymbol{h}') \tag{26}$$

for some *pseudometric* $d : \mathcal{H} \times \mathcal{H} \to \mathbb{R}_{\geq 0}$ satisfying Assumption 1 and bounded, equinumerous *multisets* $\boldsymbol{H}^{(1)}, \boldsymbol{H}^{(2)}$. By Theorem 3, the *pseudometric* $d$ has a corresponding positive definite kernel $k : \mathcal{H} \times \mathcal{H} \to \mathbb{R}$. A simple algebraic manipulation and using the fact that $\boldsymbol{H}^{(1)}$ and $\boldsymbol{H}^{(2)}$ are equinumerous results in

$$D^2\left(\boldsymbol{H}^{(1)}, \boldsymbol{H}^{(2)}\right) = \sum_{\substack{\boldsymbol{h} \in \boldsymbol{H}^{(1)} \\ \boldsymbol{h}' \in \boldsymbol{H}^{(1)}}} k(\boldsymbol{h}, \boldsymbol{h}') + \sum_{\substack{\boldsymbol{h} \in \boldsymbol{H}^{(2)} \\ \boldsymbol{h}' \in \boldsymbol{H}^{(2)}}} k(\boldsymbol{h}, \boldsymbol{h}') - 2 \sum_{\substack{\boldsymbol{h} \in \boldsymbol{H}^{(1)} \\ \boldsymbol{h}' \in \boldsymbol{H}^{(2)}}} k(\boldsymbol{h}, \boldsymbol{h}'). \tag{27}$$

Note that $D$ is indeed a *pseudometric* since $k$ is positive definite as noted by Joshi et al. (2011).[3] By the reproducing property of $k$ and the linearity of the inner product, it may be shown that

$$\left\| G\left(\boldsymbol{H}^{(1)}\right) - G\left(\boldsymbol{H}^{(2)}\right) \right\| = D\left(\boldsymbol{H}^{(1)}, \boldsymbol{H}^{(2)}\right), \tag{28}$$

where

$$G(\boldsymbol{H}) = \sum_{\boldsymbol{h} \in \boldsymbol{H}} g(\boldsymbol{h}) \tag{29}$$

and $g$ is the corresponding feature map of the kernel $k$. Hence, for every $\varepsilon_1 > \varepsilon_2 > 0$,

$$\varepsilon_2 < \left\| G\left(\boldsymbol{H}^{(1)}\right) - G\left(\boldsymbol{H}^{(2)}\right) \right\| < \varepsilon_1 \iff \varepsilon_2 < D\left(\boldsymbol{H}^{(1)}, \boldsymbol{H}^{(2)}\right) < \varepsilon_1. \tag{30}$$

$\square$

## B  Additional benchmark experiments

**Heterophilous Datasets**   Heterophilous Datasets (Platonov et al., 2023b) is a collection of node property prediction benchmark datasets for evaluating GNNs under heterophily. In particular, the roman-empire, amazon-ratings, tolokers, and questions datasets have uncountable node features. Meanwhile, the minesweeper dataset has countable node features. The performance metric of roman-empire and amazon-ratings is accuracy while the performance metric of the remaining datasets is ROC-AUC. Platonov et al. (2023b) provides more information regarding the individual datasets.

Table 5 presents the mean and standard deviation of the test performance for SIR-GCN, GCN, GraphSAGE, and GAT across the five benchmark datasets with the experimental set-up closely following that of Platonov et al. (2023b). The test performance for the GraphTransformer (Shi et al., 2021) and heterophilous GNNs—H$_2$GCN (Zhu et al., 2020), CPGNN (Zhu et al., 2021), GPR-GNN (Chien et al., 2021), FSGNN (Maurya et al., 2022), GloGNN (Li et al., 2022), FAGCN (Bo et al., 2021), GBK-GNN (Du et al., 2022), and JacobiConv (Wang & Zhang, 2022)—are also presented as additional baselines. Unsurprisingly, SIR-GCN performs poorly on amazon-ratings, tolokers, and questions as these datasets exhibit near-zero label informativeness (Platonov et al., 2023a) between key node labels and query node labels, which severely limits the ability of SIR-GCN to learn meaningful relationships between neighboring nodes. Despite this inherent challenge, the performance gap in ROC-AUC between SIR-GCN and the top-performing models on tolokers and questions remains minimal, reflecting its robustness. Conversely, SIR-GCN outperforms classical GNNs, GraphTransformer, and all heterophilous GNNs on roman-empire and minesweeper, underscoring its utility even in heterophilous settings. Overall, the results highlight that SIR-GCN remains competitive in heterophilous graph tasks. Future works may further explore its theoretical and empirical properties under heterophily.

---

[3]If $k$ is also *integrally strictly positive definite* (Sriperumbudur et al., 2010), then the hash function $G$ becomes injective and $D$ becomes a metric.

Table 5: Test performance on Heterophilous Datasets.

| Model | roman-empire ($\uparrow$) | amazon-ratings ($\uparrow$) | minesweeper ($\uparrow$) | tolokers ($\uparrow$) | questions ($\uparrow$) |
|---|---|---|---|---|---|
| GCN | $73.69 \pm 0.74$ | $48.70 \pm 0.63$ | $89.75 \pm 0.52$ | $83.64 \pm 0.67$ | $76.09 \pm 1.27$ |
| GraphSAGE | $85.74 \pm 0.67$ | $\mathbf{53.63 \pm 0.39}$ | $93.51 \pm 0.57$ | $82.43 \pm 0.44$ | $76.44 \pm 0.62$ |
| GAT | $80.87 \pm 0.30$ | $49.09 \pm 0.63$ | $92.01 \pm 0.68$ | $\mathbf{83.70 \pm 0.47}$ | $77.43 \pm 1.20$ |
| GraphTransformer | $86.51 \pm 0.73$ | $51.17 \pm 0.66$ | $91.85 \pm 0.76$ | $83.23 \pm 0.64$ | $77.95 \pm 0.68$ |
| H$_2$GCN | $60.11 \pm 0.52$ | $36.47 \pm 0.23$ | $89.71 \pm 0.31$ | $73.35 \pm 1.01$ | $63.59 \pm 1.46$ |
| CPGNN | $63.96 \pm 0.62$ | $39.79 \pm 0.77$ | $52.03 \pm 5.46$ | $73.36 \pm 1.01$ | $65.96 \pm 1.95$ |
| GPR-GNN | $64.85 \pm 0.27$ | $44.88 \pm 0.34$ | $86.24 \pm 0.61$ | $72.94 \pm 0.97$ | $55.48 \pm 0.91$ |
| FSGNN | $79.92 \pm 0.56$ | $52.74 \pm 0.83$ | $90.08 \pm 0.70$ | $82.76 \pm 0.61$ | $\mathbf{78.86 \pm 0.92}$ |
| GloGNN | $59.63 \pm 0.69$ | $36.89 \pm 0.14$ | $51.08 \pm 1.23$ | $73.39 \pm 1.17$ | $65.74 \pm 1.19$ |
| FAGCN | $65.22 \pm 0.56$ | $44.12 \pm 0.30$ | $88.17 \pm 0.73$ | $77.75 \pm 1.05$ | $77.24 \pm 1.26$ |
| GBK-GNN | $74.57 \pm 0.47$ | $45.98 \pm 0.71$ | $90.85 \pm 0.58$ | $81.01 \pm 0.67$ | $74.47 \pm 0.86$ |
| JacobiConv | $71.14 \pm 0.42$ | $43.55 \pm 0.48$ | $89.66 \pm 0.40$ | $68.66 \pm 0.65$ | $73.88 \pm 1.16$ |
| SIR-GCN | $\mathbf{87.67 \pm 0.28}$ | $46.73 \pm 0.61$ | $\mathbf{94.12 \pm 0.42}$ | $82.85 \pm 0.72$ | $75.33 \pm 1.34$ |

## C   Experimental set-up

All experiments are performed on a single NVIDIA® Quadro RTX 6000 (24GB) card using the Deep Graph Library (DGL) (Wang et al., 2019a) with PyTorch (Paszke et al., 2019) backend. For synthetic datasets, the reported results are obtained from the models at the final epoch across 10 trials with varying seed values. For benchmark datasets, the reported results are obtained from the models with the best validation loss across the 10 trials. The codes to reproduce the results may be found at `https://github.com/briangodwinlim/SIR-GCN`.

### C.1   Synthetic datasets

**DictionaryLookup**   Adopting Brody et al. (2022), the training dataset consists of 4,000 bipartite graphs, each containing $2n$ nodes with randomly assigned attributes and/or values, while the test dataset comprises 1,000 bipartite graphs with the same configuration. All models utilize a single GNN layer with $4n$ hidden units. A two-layer MLP is also used for GIN and $\sigma$ of SIR-GCN while PNA uses the sum, max, and standard deviation aggregators. Model training is performed with the AdamW (Loshchilov & Hutter, 2019) optimizer for 500 epochs with a batch size of 256 and a learning rate of 0.001 that decays by a factor of 0.5 with patience of 10 epochs based on the training loss.

**HeteroEdgeCount**   The training dataset consists of 4,000 directed graphs, each containing a maximum of 50 nodes with uniformly selected edges using the `rand_graph` function of DGL and uniformly assigned node labels from one of $c$ classes using the `randint` function of PyTorch. These measures ensure that the graphs are sufficiently diverse with respect to graph structure and heterophily. Meanwhile, the test dataset comprises 1,000 directed graphs with the same configuration. All models utilize a single GNN layer with $10c$ hidden units and sum pooling as the graph readout function. A feed-forward neural network is also used for GIN while PNA uses the sum, max, and standard deviation aggregators. Model training is performed with the AdamW optimizer for 500 epochs with a batch size of 256 and a learning rate of 0.001 that decays by a factor of 0.5 with patience of 10 epochs based on the training loss.

### C.2   Benchmark datasets

**Benchmarking GNNs**   The datasets are obtained from `dgl` with data splits (training, validation, test) following Dwivedi et al. (2023). In line with Dwivedi et al. (2023), Corso et al. (2020), and Tailor et al. (2022), all models utilize 4 GNN layers with batch normalization and residual connections while constrained to a parameter budget of 100,000. Regularization with weights in $\left\{1 \times 10^{-7}, 1 \times 10^{-6}, 1 \times 10^{-5}\right\}$ and dropouts with rates in $\{0.1, 0.2, 0.3\}$ are also used to prevent overfitting. The mean, symmetric mean, and max aggregators are used since the sum aggregator is observed to not generalize well to unseen graphs as noted

by Veličković et al. (2020). Additionally, sum pooling is used as the graph readout function for ZINC while mean pooling is used for MNIST and CIFAR10. Model training is performed with the AdamW optimizer for a maximum of 500 epochs with a batch size of 128, whenever applicable, and a learning rate of 0.001 that decays by a factor of 0.5 with patience of 10 epochs based on the training loss. The reported results for the other models in Table 3 are obtained from Dwivedi et al. (2023), Corso et al. (2020), and Tailor et al. (2022).

**Open Graph Benchmark**  The datasets are obtained from `ogb`, the Open Graph Benchmark Python package, with data splits following Hu et al. (2020a). In line with Corso et al. (2020) and Tailor et al. (2022), the model for ogbn-arxiv and ogbg-molhiv utilizes 3 and 4 GNN layers, respectively, with batch normalization and residual connections while constrained to a parameter budget of 100,000. Weight decays in $\{1 \times 10^{-4}, 1 \times 10^{-3}\}$ and dropouts with a rate of 0.2 are also used to prevent overfitting. The symmetric mean and max aggregators are used with mean pooling as the graph readout function for ogbg-molhiv. Model training is performed with the AdamW optimizer for a maximum of 1000 and 100 epochs, respectively, with a batch size of 64 for ogbg-molhiv and learning rates in $\{0.001, 0.01\}$ that decays by a factor of 0.5 with patience of 40 and 10 epochs, respectively, based on the training loss. The reported results for the other models in Table 4 are obtained from Corso et al. (2020) and Tailor et al. (2022).

**Heterophilous Datasets**  The datasets are obtained from `dgl` with data splits following Platonov et al. (2023b). In line with Platonov et al. (2023b), the number of GNN layers is chosen from $\{3, 5\}$, employing residual connections and dropouts with a rate of 0.2. Moreover, the hidden dimension is chosen from $\{256, 512\}$, utilizing batch and layer normalization as well as the symmetric mean and max aggregators. Model training is performed with the AdamW optimizer for a maximum of 1000 epochs and a learning rate of $3 \times 10^{-5}$. The reported results for the other models in Table 5 are obtained from Platonov et al. (2023b).

## D   Runtime analysis

As an additional evaluation, the inference runtime for each model in the synthetic datasets is presented in Tables 6 and 7. The results, when considered alongside Tables 1 and 2, illustrate that SIR-GCN achieves a balance between computational complexity and model expressivity, specifically with regards to PNA which is also designed for uncountable node features.

Table 6: DictionaryLookup inference runtime.

| Model | $n = 10$ | $n = 20$ | $n = 30$ | $n = 40$ | $n = 50$ |
|---|---|---|---|---|---|
| GCN | 0.3526s ± 0.0778s | 0.4734s ± 0.0468s | 0.4777s ± 0.0854s | 0.5619s ± 0.0518s | 0.5520s ± 0.0679s |
| GraphSAGE | 0.4565s ± 0.0873s | 0.5264s ± 0.0317s | 0.5716s ± 0.1132s | 0.7742s ± 0.0597s | 0.9193s ± 0.0473s |
| GATv2 | 0.3950s ± 0.1017s | 0.5276s ± 0.0556s | 0.6191s ± 0.0879s | 0.7472s ± 0.0346s | 1.0065s ± 0.0280s |
| GIN | 0.3696s ± 0.0899s | 0.4610s ± 0.0459s | 0.4670s ± 0.0781s | 0.5947s ± 0.0548s | 0.5194s ± 0.0993s |
| PNA | 0.8854s ± 0.0412s | 1.1913s ± 0.1024s | 1.4526s ± 0.0684s | 1.8793s ± 0.0528s | 2.8387s ± 0.0603s |
| SIR-GCN | 0.4687s ± 0.0777s | 0.6066s ± 0.0398s | 0.8053s ± 0.0485s | 1.1496s ± 0.0427s | 1.7031s ± 0.0458s |

Table 7: HeteroEdgeCount inference runtime.

| Model | $c = 2$ | $c = 4$ | $c = 6$ | $c = 8$ | $c = 10$ |
|---|---|---|---|---|---|
| GCN | 0.4243s ± 0.0520s | 0.3852s ± 0.0517s | 0.3868s ± 0.0743s | 0.4166s ± 0.0551s | 0.4177s ± 0.0494s |
| GraphSAGE | 0.4691s ± 0.0400s | 0.4790s ± 0.0440s | 0.4399s ± 0.0629s | 0.4501s ± 0.0603s | 0.4964s ± 0.0601s |
| GATv2 | 0.4710s ± 0.0978s | 0.4941s ± 0.0567s | 0.4718s ± 0.0361s | 0.5514s ± 0.0608s | 0.5437s ± 0.0724s |
| GIN | 0.4085s ± 0.0741s | 0.3875s ± 0.0627s | 0.3855s ± 0.0645s | 0.4298s ± 0.0566s | 0.4329s ± 0.0534s |
| PNA | 2.2963s ± 0.0413s | 2.4238s ± 0.0611s | 2.4577s ± 0.0533s | 2.4741s ± 0.0665s | 2.5623s ± 0.0425s |
| SIR-GCN | 0.5338s ± 0.0353s | 0.5264s ± 0.0737s | 0.5635s ± 0.0695s | 0.5764s ± 0.0401s | 0.6230s ± 0.0388s |

Table 8 further complements these results by presenting the asymptotic computational runtime complexity of the different GNNs. In particular, SIR-GCN first computes the linear transformations $\boldsymbol{W_Q h_u}$ and $\boldsymbol{W_K h_v}$ for every node which incurs $\mathcal{O}\left(|\mathcal{V}| \times d_{\text{hidden}} \times d_{\text{in}}\right)$. Afterward, $\sigma\left(\boldsymbol{W_Q h_u} + \boldsymbol{W_K h_v}\right)$ is computed for every edge, using the previously calculated values, and aggregated across the neighbors of each node which incurs $\mathcal{O}\left(|\mathcal{E}| \times d_{\text{hidden}}\right)$. Finally, the aggregated values are linearly transformed with $\boldsymbol{W_R}$ for every node which incurs $\mathcal{O}\left(|\mathcal{V}| \times d_{\text{out}} \times d_{\text{hidden}}\right)^4$. In total, since SIR-GCN employs only linear transformations along nodes and only an activation function along edges, its computational complexity is comparable to GCN, GraphSAGE, GAT, GATv2, and GIN. Specifically, these models achieve computational efficiency by maintaining linear complexity along edges attributed to activation functions and neighborhood aggregation. Despite this, SIR-GCN consistently outperforms these classical GNNs across all benchmarks. Notably, SIR-GCN also demonstrates a lower complexity than PNA due to the number of aggregators used, yet delivers superior performance across *most* datasets. These additional analyses further underscore the practical utility of the proposed model.

Table 8: Asymptotic runtime complexity.

| Model | Complexity |
|---|---|
| GCN | $\mathcal{O}\left(|\mathcal{V}| \times d_{\text{out}} \times d_{\text{in}} + |\mathcal{E}| \times d_{\text{out}}\right)$ |
| GraphSAGE | $\mathcal{O}\left(|\mathcal{V}| \times d_{\text{out}} \times d_{\text{in}} + |\mathcal{E}| \times d_{\text{out}}\right)$ |
| GAT/GATv2 | $\mathcal{O}\left(|\mathcal{V}| \times d_{\text{out}} \times d_{\text{in}} + |\mathcal{E}| \times d_{\text{out}}\right)$ |
| GIN | $\mathcal{O}\left(|\mathcal{E}| \times d_{\text{in}} + |\mathcal{V}| \times \texttt{MLP}\right)$ |
| PNA | $\mathcal{O}\left(|\mathcal{E}| \times d_{\text{in}}^2 + |\mathcal{E}| \times d_{\text{in}} \times k + |\mathcal{V}| \times d_{\text{out}} \times d_{\text{in}} \times k\right)$ |
| SIR-GCN | $\mathcal{O}\left(|\mathcal{V}| \times d_{\text{hidden}} \times d_{\text{in}} + |\mathcal{E}| \times d_{\text{hidden}} + |\mathcal{V}| \times d_{\text{out}} \times d_{\text{hidden}}\right)$ |

Note: $k$ represents the number of aggregators and scalers in PNA.

# E SIR-GCN extensions

Denote $\boldsymbol{h_{u,v}}$ as the feature of the edge connecting node $v$ to node $u$. Following the intuition presented in Section 3.1, SIR-GCN with residual connection may be modified to leverage edge features to obtain

$$\boldsymbol{h_u^*} = \text{MLP}_{\text{Res}}(\boldsymbol{h_u}) + \sum_{v \in \mathcal{N}(u)} \boldsymbol{W_R}\,\sigma\left(\boldsymbol{W_Q h_u} + \boldsymbol{W_E h_{u,v}} + \boldsymbol{W_K h_v}\right), \tag{31}$$

where $\boldsymbol{W_E} \in \mathbb{R}^{d_{\text{hidden}} \times d_{\text{in}}}$. Consequently, this also increases the computational complexity of the model to

$$\mathcal{O}\left(|\mathcal{E}| \times d_{\text{hidden}} \times d_{\text{in}} + |\mathcal{V}| \times d_{\text{out}} \times d_{\text{hidden}} + |\mathcal{V}| \times \texttt{MLP}_{\text{Res}}\right), \tag{32}$$

where $\texttt{MLP}_{\text{Res}}$ denotes the computational complexity of $\text{MLP}_{\text{Res}}$, making it comparable to PNA. Similarly, this extension may be viewed as a generalization of GIN with edge features (Hu et al., 2020b).

Furthermore, one may also inject inductive bias into the *pseudometrics* which may correspond to specifying the architecture type for the corresponding message function $g$. For instance, if node features are known to have a sequential relationship (*e.g.*, stock (Hsu et al., 2023) and fMRI (Kim & Ye, 2020) data), $g$ may then be aptly modeled using recurrent or convolutional networks.

---

[4] In the case of SIR-GCN with max aggregation, the linear transformation and the max aggregator cannot be interchanged. Hence, the linear transformation $\boldsymbol{W_R}$ must be performed along edges which incurs $\mathcal{O}\left(|\mathcal{E}| \times d_{\text{out}} \times d_{\text{hidden}}\right)$.

