# OpenReview forum: "Contextualized Messages Boost Graph Representations"
_TMLR — Accepted by TMLR_

### Review · Reviewer_ypW5 · 2025-02-21

**Summary Of Contributions:**

The paper proposes the Soft-Isomorphic Relational Graph Convolution Network (SIR-GCN), for understanding the representational capabilities of GNNs when the node feature space is uncountable.
Specifically, it introduces the concept of soft-injective functions by employing pseudometric distances, allowing distinct inputs to produce similar outputs only if they are deemed sufficiently similar by the pseudometric. This relaxes the strict injective and metric requirements in previous GNN models.
The proposed SIR-GCN performs the contextualized transformation of neighborhood feature representations using anisotropic and dynamic message functions by modeling the anisotropic and dynamic soft-injective relational message passing with a two-layer MLP that can implicitly learns the pseudometrics.
The authors provide a mathematical analysis comparing SIR-GCN with well-known GNNs such as GCN, GraphSAGE, GAT, and PNA, showing that other GNNs can be a special case of SIR-GCN.

**Audience:**

Yes

**Broader Impact Concerns:**

The authors show that the proposed GNN architecture outperforms in predicting molecular properties, which can be maliciously used to select toxic compounds.

**Claims And Evidence:**

Yes

**Requested Changes:**

Minor Request
* It would be better to relate the ability to model the distinct inputs to Figure 1, to emphasize the importance of the proposed concepts.
* It is promising to have the comparable computational complexity from existing GNNs. However, the in-depth comparison should be provided to enhance the readability.

**Strengths And Weaknesses:**

Strengths
* The paper provides a mathematical analysis comparing SIR-GCN with well-known GNNs, establishing a theoretical basis of why the proposed method outperforms the existing GNNs.
* The experimental results show that the proposed method outperforms existing GNNs.
* By employing pseudometric distances and soft-injective functions, the proposed method relaxes strict injective and metric requirements, allowing the model to handle distinct inputs more accurately.

Weaknesses
* The performance gain could come from the additional parameters such as MLPs, compared to the baselines.
* Even though the proposed SIR-GCN outperforms the existing GNNs, I wonder if the proposed GNN can outperform the recent powerful models such as graph transformers. Otherwise, it would be better to discuss whether learning the pseudometric distances could be possible in attention-based architectures.

---

> ### Author Response · Authors · 2025-02-26
>
> We thank the reviewer for the positive feedback.
>
> **Weakness 1.** In Table 3, all models, including SIR-GCN, are constrained to a parameter budget of 100,000 following Dwivedi et al. (2023). This ensures a fair evaluation wherein performance differences are solely attributed to the GNN architectural design and **not** due to the additional parameters. This is also stated in Appendix C2.
>
> **Weakness 2.** Since SIR-GCN is based on the classical message-passing framework for GNNs, its performance without employing additional tricks remains limited when compared against state-of-the-art models, such as graph transformers. In particular, the *pseudometric* $D$ on bounded, equinumerous *multisets* of $\mathcal{H}$ considered in Corollary 1 computes the distance between two *multisets* by comparing their elements in a pair-wise manner. This formulation does not inherently possess an attention mechanism that would allow comparing elements with the context of the other elements in the *multisets*. In the Conclusion section, we have added a recommendation for future works to consider more complex *pseudometric* formulations. This may include equipping the *pseudometric* with an attention mechanism that would result in an aggregation function comparable to graph transformers.
>
> **Request 1.** We agree with the reviewer that Figure 1 may be improved to better understand the importance of the proposed concepts. In response, we have modified Figure 1 to provide a simple illustration of *soft-injective* functions mapping distinct elements of $\mathcal{H}$ to the same point in $\mathcal{S}$.
>
> **Request 2.** We agree with the reviewer that additional discussions on the computational complexity would help improve readability. In response, we have provided a more detailed analysis of the asymptotic computational runtime complexity in Appendix D.
>
> **Broader Impact Concerns.** While we acknowledge the concern of the reviewer that SIR-GCN may be maliciously used to select toxic compounds, we believe that the risks are not substantially different from those of other GNN models.
>
>
> **References**
>
> Vijay Prakash Dwivedi, Chaitanya K. Joshi, Anh Tuan Luu, Thomas Laurent, Yoshua Bengio, and Xavier Bresson. Benchmarking graph neural networks. Journal of Machine Learning Research, 24(43):1-48, 2023.

---

### Review · Reviewer_wxBX · 2025-02-21

**Summary Of Contributions:**

This paper studies GNNs with uncountable node feature representation. It relaxed the requirements for injection and metrics when designing the aggregation function. The key idea was to define a pseudometric distance in a soft-injective function, such that the inputs result in similar outputs if the distance between them is sufficiently small on some representation. A SIR-GCN framework was then proposed based on the anisotropic and dynamic message functions.

**Audience:**

Yes

**Claims And Evidence:**

Yes

**Requested Changes:**

The GraphHeterophily data set can be further explained to discuss the impact of graph heterophily on SIR-GCN based on the anisotropic and dynamic message functions. More heterophily-based GNNs can be considered to validate the effectiveness of SIR-GCN on GraphHeterophily.

**Strengths And Weaknesses:**

Strengths:
(S1) This work highlighted the soft-injective function in understanding the representation capability of GNNs with uncountable node feature representation.

(S2) The proposed SIR-GCN was simple and computationally efficient. The connections between SIR-GCN and existing GNNs were discussed

(S3) Experimental results demonstrated the superior performance of SIR-GCN over baselines.

Weaknesses:
(W1) Figure 2 is not well-explained. It is unclear regarding the MLP-induced message functions.

(W2) More recent GNN baselines can be included in the experiments.

---

> ### Author Response · Authors · 2025-02-26
>
> We thank the reviewer for the insightful feedback.
>
> **Weakness 1.** In the section on *dynamic* transformation, we show that for arbitrary *pseudometrics*, the corresponding message function $g_u$ is not necessarily linear and possibly more complex. In practice, this highlights the significance of *dynamic* (*i.e.*, a universal function approximator) message functions $g_u$ in the MPNN framework, which may be modeled as multi-layer perceptrons (MLPs) as illustrated in Figs. 2b and 2c.
>
> **Weakness 2.** We have provided additional results for GATv2 and the more recent efficient graph convolution (EGC-S and EGC-M) (Tailor et al., 2022) in Table 3. These serve as additional MPNN-based baselines. The results underscore that, under the same constraints, SIR-GCN consistently outperforms MPNN-based baselines despite its simplicity, establishing it as a promising alternative to existing MPNNs.
>
> **Request 1.** We have renamed the GraphHeterophily dataset to HeteroEdgeCount for clarity as the task simply involves counting the number of heterophilous edges in each graph rather than evaluating the impact of heterophily on GNN performance. This synthetic dataset is specifically designed to illustrate a **trivial task involving countable node features where even the theoretically powerful models like GIN and PNA fail**. While it demonstrates the utility of *anisotropic* and *dynamic* message functions in learning the relationships between neighboring nodes, **it does not claim that these features inherently enhance performance in heterophilous graphs**, as **evaluating GNNs under heterophily is not the purpose of this dataset**. Moreover, since **heterophilous graphs fall beyond the scope of this work** and **SIR-GCN is *not* explicitly designed for heterophily**, we do not compare it with heterophilous GNNs (similar to how GIN and PNA are not compared against heterophilous GNNs). Nonetheless, we have added a brief remark in Section 4.6 discussing how *anisotropic* and *dynamic* message functions may potentially aid the non-uniform neighborhood aggregation in heterophilous graphs (Zheng et al., 2024). A formal analysis of this relationship is left for future work.
>
> Furthermore, we note that heterophilous GNNs largely build upon GCN and GraphSAGE by integrating various techniques to combine multi-hop neighborhood information. However, in the context of HeteroEdgeCount, this feature becomes irrelevant as the necessary information lies solely within the 1-hop neighborhood. Additionally, since these heterophilous GNNs are fundamentally extensions of GCN and GraphSAGE, they also inherit their limitations—namely, *static* and *isotropic* message functions—which we have demonstrated to hinder performance on this synthetic dataset. To further illustrate this point, we have provided the test MSE for heterophilous GNNs—FSGNN (Maurya et al., 2022), FAGCN (Bo et al., 2021), and GBK-GNN (Du et al., 2022)—below. These results complement our above observations and highlight the utility of the *anisotropic* and *dynamic* message function in SIR-GCN.
>
> |  Model  |   $c = 2$   |   $c = 4$   |   $c = 6$   |   $c = 8$   |   $c = 10$   |
> | :-----: | :------------: | :------------: | :------------: | :------------: | :------------: |
> |  FSGNN  | 16030 ± 1108 | 26798 ± 2239 | 27742 ± 2112 | 25360 ± 2323 | 23921 ± 2235 |
> |  FAGCN  | 23858 ± 1267 | 53121 ± 3011 | 65515 ± 3453 | 72352 ± 3859 | 76495 ± 4169 |
> | GBK-GNN | 23700 ± 1430 | 52585 ± 2692 | 64531 ± 3676 | 70911 ± 3849 | 74702 ± 4691 |
> | SIR-GCN | 0.001 ± 0.000 | 0.004 ± 0.005 | 1.495 ± 4.428 | 0.038 ± 0.068 | 0.089 ± 0.134 |
>
>
> **References**
>
> Deyu Bo, Xiao Wang, Chuan Shi, and Huawei Shen. Beyond low-frequency information in graph convolutional networks. Proceedings of the AAAI Conference on Artificial Intelligence, 35(5):3950–3957, 2021.
>
> Lun Du, Xiaozhou Shi, Qiang Fu, Xiaojun Ma, Hengyu Liu, Shi Han, and Dongmei Zhang. GBK-GNN: Gated bi-kernel graph neural networks for modeling both homophily and heterophily. In Proceedings of the ACM Web Conference 2022, WWW ’22, pp. 1550–1558. Association for Computing Machinery, 2022.
>
> Sunil Kumar Maurya, Xin Liu, and Tsuyoshi Murata. Simplifying approach to node classification in graph neural networks. Journal of Computational Science, 62:101695, 2022.
>
> Shyam A. Tailor, Felix Opolka, Pietro Lio, and Nicholas Donald Lane. Do we need anisotropic graph neural networks? In International Conference on Learning Representations, 2022.
>
> Xin Zheng, Yi Wang, Yixin Liu, Ming Li, Miao Zhang, Di Jin, Philip S. Yu, and Shirui Pan. Graph neural networks for graphs with heterophily: A survey, 2024. arXiv:2202.07082.

---

### Review · Reviewer_u4dh · 2025-02-22

**Summary Of Contributions:**

This paper improves the existing theoretical framework for GNN expressiveness in the case of uncountable node feature representation. It proposes a new message-passing aggregation method within this framework. This method can be seen as a generalization of classical GNN approaches. The results on synthetic and real-world datasets show that the proposed method outperforms simple yet classic GNN methods.

**Audience:**

Yes

**Broader Impact Concerns:**

None.

**Claims And Evidence:**

Yes

**Requested Changes:**

Please refer to the weaknesses.

**Strengths And Weaknesses:**

### Strengths
- This paper is well-written and easy to follow.
- The case of uncountable node feature representation mentioned in this paper is currently less explored in the community.

### Weaknesses
- The paper repeatedly mentions the term "metric" but does not provide a definition. The authors should add more information to make the paper more accessible to readers with different research backgrounds, such as why it is necessary to define "metric" in the context of the research problem and its role within the existing theoretical framework.
- The soft-injective message function proposed in the paper has two properties, i.e., dynamic transformation and anisotropic messages. The authors should provide further explanation to clarify what "dynamic" transformation means, why it is important, and what type of function corresponds to a static transformation.
- The authors spend a considerable amount of space in Section 4 discussing the relationship between the proposed method SIR-GCN and various classical methods. This section is not very informative, and it is recommended to present the comparison directly in a table or other concise formats.
- In R1, the authors prove that by tuning hyperparameters, classical GNN methods can also achieve competitive performance on heterophilic graphs. The authors should conduct experiments on more real-world datasets (e.g., larger graphs, real-world homophilic and heterophilic graphs, R2-R4) and compare the results with those in R1.

R1. Classic GNNs are Strong Baselines. NeurIPS'24

R2. Large Scale Learning on Non-Homophilous Graphs: New Benchmarks and Strong Simple Methods. NeurIPS' 21

R3. A critical look at the evaluation of GNNs under heterophily. ICLR' 23

R4. Open Graph Benchmark: Datasets for Machine Learning on Graphs.

---

> ### Author Response · Authors · 2025-02-26
>
> We thank the reviewer for the constructive feedback.
>
> **Weakness 1.** We agree with the reviewer that adding more information regarding metrics may help readers better understand their role within the context of the paper. Specifically, we added more explanations on how metrics and injective functions ensure a unique mapping in the embedded feature space, which is crucial for tasks requiring graph isomorphism. To strengthen the discussion, we have included a brief mathematical definition of a metric, where the first condition of Definition 1 is replaced with $d\left(\boldsymbol{h}^{(1)}, \boldsymbol{h}^{(2)}\right) = 0 \Longleftrightarrow \boldsymbol{h}^{(1)} = \boldsymbol{h}^{(2)}$, ensuring points in $\mathcal{H}$ are distinguishable and unique with respect to $d$. We contrast this with a *pseudometric* $d$, where $d\left(\boldsymbol{h}^{(1)}, \boldsymbol{h}^{(2)}\right) = 0$ does not necessarily imply that $\boldsymbol{h}^{(1)} = \boldsymbol{h}^{(2)}$, highlighting how our proposed theoretical framework relaxes the distinguishability constraint of a metric implicitly employed in previous works that rely on injectivity.
>
> **Weakness 2.** We agree with the reviewer that providing further clarifications on *dynamic* functions may improve readability. In the section on *dynamic* transformation, we show that for arbitrary *pseudometrics*, the corresponding message function $g_u$ is not necessarily linear and possibly more complex. In practice, this highlights the significance of *dynamic* (*i.e.*, a universal function approximator) message functions $g_u$ in the MPNN framework, which may be modeled as MLPs, to provide sufficient expressivity to approximate arbitrary and complex message transformations. We contrast this to *static* message functions or function approximators with limited expressivity, such as linear transformations and perceptrons.
>
> **Weakness 3.** We respectfully disagree with the assessment that the mathematical discussions in Section 4 are not informative, as they serve a critical role in **contextualizing SIR-GCN within the broader GNN literature**. This is further supported by the evaluations of Reviewers wxBX and ypW5. The section provides a rigorous theoretical foundation, demonstrating how **SIR-GCN generalizes classical GNNs** and offering insights into **why it consistently outperforms these models** in the experimental results. These also motivate **SIR-GCN as a promising candidate for practical GNN applications**. Simply presenting these comparisons in a tabular format would significantly undermine the purpose of the section. Nevertheless, we acknowledge the importance of clarity and conciseness and remain open to specific suggestions for streamlining the discussion while preserving its rigor.
>
> **Weakness 4.** We thank the reviewer for directing us to the *contemporaneous and recent work* of Luo et al. (2024). Upon closer reading of their work, we noted that their primary focus and contribution is on optimizing classical GNNs through ***extensive hyperparameter tuning*, requiring *significant computational resources* and covering nearly *3,000 configurations* for each model**, to achieve state-of-the-art (SOTA) performance specifically on **node classification datasets**. In contrast, our work focuses on **diverse node and graph property prediction datasets** and follows the **established experimental set-up of Dwivedi et al. (2023), Corso et al. (2020), and Tailor et al. (2022) to prioritize fair evaluation** over achieving SOTA by ensuring **consistency in model architecture and parameter count**, thereby isolating the effect of GNN architectural design from performance differences attributed to additional parameters or architectural tuning. Within this controlled setting, SIR-GCN consistently outperforms MPNN-based baselines, demonstrating its effectiveness despite its simplicity. While hyperparameter tuning may further enhance the performance of SIR-GCN, such optimizations are orthogonal to the objectives of the current work and are best explored in future research.
>
> Furthermore, **SIR-GCN is explicitly designed for tasks involving uncountable node features**. In this regard, the six benchmark datasets already **encompass a diverse range of GNN tasks**, enabling a robust and comprehensive evaluation of model performance. Nevertheless, we have provided **additional experimental results for the large-scale datasets from Hu et al. (2020)** following the reviewer's request. Crucially, we make **no claims regarding the performance of SIR-GCN on heterophilous graphs, as it is not explicitly designed for such graphs**. Consequently, we also do not compare SIR-GCN with heterophilous GNNs on heterophilous datasets (similar to how GIN and PNA are not evaluated on heterophilous graphs). A formal analysis of how *anisotropic* and *dynamic* message functions influence performance on heterophilous graphs is an important avenue for future exploration.

---

> ### Author Response · Authors · 2025-02-26
>
> **References**
>
> Gabriele Corso, Luca Cavalleri, Dominique Beaini, Pietro Liò, and Petar Veličković. Principal neighbourhood aggregation for graph nets. In Advances in Neural Information Processing Systems, volume 33, pp.13260-13271, 2020.
>
> Vijay Prakash Dwivedi, Chaitanya K. Joshi, Anh Tuan Luu, Thomas Laurent, Yoshua Bengio, and Xavier Bresson. Benchmarking graph neural networks. Journal of Machine Learning Research, 24(43):1-48, 2023.
>
> Weihua Hu, Matthias Fey, Marinka Zitnik, Yuxiao Dong, Hongyu Ren, Bowen Liu, Michele Catasta, and Jure Leskovec. Open graph benchmark: Datasets for machine learning on graphs. In Advances in Neural Information Processing Systems, volume 33, pp. 22118–22133, 2020.
>
> Yuankai Luo, Lei Shi, and Xiao-Ming Wu. Classic GNNs are strong baselines: Reassessing GNNs for node classification. In The Thirty-eight Conference on Neural Information Processing Systems Datasets and Benchmarks Track, 2024.
>
> Shyam A. Tailor, Felix Opolka, Pietro Lio, and Nicholas Donald Lane. Do we need anisotropic graph neural networks? In International Conference on Learning Representations, 2022.

---

> ### Author Response · Authors · 2025-03-08
> **Additional Experimental Results**
>
> Following the reviewer's request, we have provided additional experimental results for the large-scale datasets (ogbn-arxiv and ogbg-molhiv) from the Open Graph Benchmark (Hu et al., 2020). These additional results highlight the significance of contextualized messages in enhancing GNN expressivity and the utility of *softly* relaxing the injective and metric requirements within the MPNN framework for *most* practical GNN applications. We believe that the 8 benchmark datasets considered in the paper already provide sufficient empirical evidence on the performance of SIR-GCN relative to the key GNNs in literature. Nevertheless, while we reiterate that SIR-GCN is **not explicitly designed for heterophilous graph tasks**, we have provided additional experimental results for the five heterophilous datasets from Platonov et al. (2023) in Appendix B and recommend its formal theoretical and empirical analyses for future works.
>
>
> **References**
>
> Weihua Hu, Matthias Fey, Marinka Zitnik, Yuxiao Dong, Hongyu Ren, Bowen Liu, Michele Catasta, and Jure Leskovec. Open graph benchmark: Datasets for machine learning on graphs. In Advances in Neural Information Processing Systems, volume 33, pp. 22118–22133, 2020.
>
> Oleg Platonov, Denis Kuznedelev, Michael Diskin, Artem Babenko, and Liudmila Prokhorenkova. A critical look at the evaluation of GNNs under heterophily: Are we really making progress? In International Conference on Learning Representations, 2023.

---

### Author Response · Authors · 2025-02-26
**Additional Clarifications**

We would like to provide further clarifications on the scope and contributions of our work.

1. The HeteroEdgeCount dataset is specifically designed to illustrate a trivial task involving countable node features where even the theoretically powerful models like GIN and PNA fail. This dataset serves to highlight the utility of *anisotropic* and *dynamic* message functions in learning the relationships between neighboring nodes. However, it **does not claim that these features inherently enhance performance in heterophilous graphs**, as evaluating GNNs on heterophily is **not the intended purpose of this dataset**.

2. SIR-GCN is **explicitly designed to handle uncountable node features**. Crucially, we **do not make any claims regarding its performance on heterophilous graphs**. Since it is not explicitly designed for such tasks, we do not compare its performance against heterophilous GNNs on heterophilous datasets (similar to how GIN and PNA are not evaluated on heterophilous graphs). While it is an interesting direction to analyze the relationship between *anisotropic* and *dynamic* message functions and heterophily, we leave its formal analysis for future work as this is **not the primary focus of the current work**.

3. We would like to highlight that Luo et al. (2024) provide results for **only a subset of our baselines** (GCN, GraphSAGE, and GAT) **only on node property prediction datasets**. Consequently, conducting extensive tuning for our other baselines (GATv2, GIN, and PNA) would be required. Furthermore, since our paper also focuses on graph property prediction datasets, extensive tuning would be necessary for all of our baselines on these new datasets. Overall, these hyperparameter tuning experiments would require **significant computational resources to cover nearly 3,000 configurations for each model**. As this presents a distinct challenge, it is best left for future works and treated in a separate paper, as our paper primarily focuses on introducing SIR-GCN.

4. We would like to explicitly clarify that our paper **does not claim that SIR-GCN achieves state-of-the-art (SOTA) performance**. Accordingly, our experimental results do not focus on attaining SOTA through an extensive hyperparameter tuning of SIR-GCN. Instead, we **prioritize a fair and unbiased evaluation**. Specifically, we closely adhere to the **established experimental set-ups** of Dwivedi et al. (2023), Corso et al. (2020), and Tailor et al. (2022), **maintaining consistency in parameter count and model architecture**. This ensures that the performance differences are solely attributed to the GNN architectural design rather than from the additional parameters or architectural tuning. Under these *controlled conditions*, SIR-GCN consistently outperforms MPNN-based baselines, establishing it as a promising alternative to existing MPNNs. Hence, **our current experimental results remain valid and underscore how SIR-GCN is a strong candidate for practical GNN applications** where the parameter budget is critical.

Overall, we believe that the claims of the paper are backed by sufficient theoretical justification and empirical evidence. We are open to further clarifications and recommendations to improve the paper.


**References**

Gabriele Corso, Luca Cavalleri, Dominique Beaini, Pietro Liò, and Petar Veličković. Principal neighbourhood aggregation for graph nets. In Advances in Neural Information Processing Systems, volume 33, pp.13260-13271, 2020.

Vijay Prakash Dwivedi, Chaitanya K. Joshi, Anh Tuan Luu, Thomas Laurent, Yoshua Bengio, and Xavier Bresson. Benchmarking graph neural networks. Journal of Machine Learning Research, 24(43):1-48, 2023.

Yuankai Luo, Lei Shi, and Xiao-Ming Wu. Classic GNNs are strong baselines: Reassessing GNNs for node classification. In The Thirty-eight Conference on Neural Information Processing Systems Datasets and Benchmarks Track, 2024.

Shyam A. Tailor, Felix Opolka, Pietro Lio, and Nicholas Donald Lane. Do we need anisotropic graph neural networks? In International Conference on Learning Representations, 2022.

---

### Decision · Action_Editor_zDwg · 2025-04-03

**Recommendation:** Accept as is

**Comment:**

There is an overall consensus that the paper should be accepted. The authors provided their source code, and it looks thorough.

**Audience:**

All reviewers agree that this paper is in line with the audience of TMLR.

**Claims And Evidence:**

There is a consensus among all reviewers that after thorough revisions, the claims are sufficiently supported by the evidence in the paper.